# NEURAL NETWORKS ARE *a priori* BIASED TOWARDS BOOLEAN FUNCTIONS WITH LOW ENTROPY

## ABSTRACT

Understanding the inductive bias of neural networks is critical to explaining their ability to generalise. Here, for one of the simplest neural networks – a single-layer perceptron with $n$ input neurons, one output neuron, and no threshold bias term – we prove that upon random initialisation of weights, the *a priori* probability $P(t)$ that it represents a Boolean function that classifies $t$ points in $\{0,1\}^n$ as 1 has a remarkably simple form: $P(t) = 2^{-n}$ for $0 \leq t < 2^n$.

Since a perceptron can express far fewer Boolean functions with small or large values of $t$ (low "entropy") than with intermediate values of $t$ (high "entropy") there is, *on average*, a strong intrinsic *a-priori* bias towards individual functions with low entropy. Furthermore, within a class of functions with fixed $t$, we often observe a further intrinsic bias towards functions of lower complexity. Finally, we prove that, regardless of the distribution of inputs, the bias towards low entropy becomes monotonically stronger upon adding ReLU layers, and empirically show that increasing the variance of the bias term has a similar effect.

## 1 INTRODUCTION

In order to generalise beyond training data, learning algorithms need some sort of inductive bias. The particular form of the inductive bias dictates the performance of the algorithm. For one of the most important machine learning techniques, deep neural networks (DNNs) (LeCun et al., 2015), sources of inductive bias can include the architecture of the networks, e.g. the number of layers, how they are connected, say as a fully connected network (FCN) or as a convolutional neural net (CNN), and the type of optimisation algorithm used, e.g. stochastic gradient descent (SGD) versus full gradient descent (GD). Many further methods such as dropout (Srivastava et al., 2014), weight decay (Krogh & Hertz, 1992) and early stopping (Morgan & Bourlard, 1990) have been proposed as techniques to improve the inductive bias towards desired solutions that generalise well. What is particularly surprising about DNNs is that they are highly expressive and work well in the heavily overparameterised regime where traditional learning theory would predict poor generalisation due to overfitting (Zhang et al., 2016). DNNs must therefore have a strong intrinsic bias that allows for good generalisation, in spite of being in the overparameterised regime.

Here we study the intrinsic bias of the parameter-function map for neural networks, defined in (Valle-Pérez et al., 2018) as the map between a set of parameters and the function that the neural network represents. In particular, we define the *a-priori* probability $P(f)$ of a DNN as the probability that a particular function $f$ is produced upon random sampling (or initialisation) of the weight and threshold bias parameters. The prior at initialization, $P(f)$, should inform the inductive bias of SGD-trained neural networks, as long as SGD approximates Bayesian inference with $P(f)$ as prior sufficiently well Valle-Pérez et al. (2018). We explain this connection further, and give some evidence supporting this behavior of SGD, in Appendix L. This supports the idea studying neural networks with random parameters Poole et al. (2016); Lee et al. (2018); Schoenholz et al. (2017); Garriga-Alonso et al. (2018); Novak et al. (2018) is not just relevant to find good initializations for optimization, but also to understand their generalization.

A naive null-model for $P(f)$ might suggest that without further information, one should expect that all functions are equally likely. However, recent very general arguments (Dingle et al., 2018) based on the coding theorem from Algorithmic Information Theory (AIT) (Li et al., 2008) have instead suggested that for a wide range of maps $M$ that obey a number of conditions such as being simple

(they have a low Kolmogorov complexity $K(M)$) and redundancy (multiple inputs map to the same output) then *if they are sufficiently biased*, they will be exponentially biased towards outputs of low Kolmogorov complexity. The parameter-function map of neural networks satisfies these conditions, and it was found empirically (Valle-Pérez et al., 2018) that, as predicted in (Dingle et al., 2018), the probability $P(f)$ of obtaining a function $f$ upon random sampling of parameter weights satisfies the following *simplicity-bias* bound

$$P(f) \lesssim 2^{-(b\widetilde{K}(f)+a)}, \tag{1}$$

where $\widetilde{K}(f)$ is a computable approximation of the true Kolmogorov complexity $K(f)$, and $a$ and $b$ are constants that depend on the network, but not on the functions.

It is widely expected that real world data is highly structured, and so has a relatively low Kolmogorov complexity (Hinton & Van Camp, 1993; Schmidhuber, 1997). The simplicity bias described above may therefore be an important source of the inductive bias that allows DNNs to generalise so well (and not overfit) in the highly over-parameterised regime (Valle-Pérez et al., 2018).

Nevertheless, this bound has limitations. Firstly, the only rigorously proven result is for the true Kolmogorov complexity version of the bound in the case of large enough $K(f)$. Although it has been found to work remarkably well for small systems and computable approximations to Kolmogorov complexity (Valle-Pérez et al., 2018; Dingle et al., 2018), this success is not yet fully understood theoretically. Secondly, it does not explain why models like DNNs are biased; it only explains that, if they are biased, they should be biased towards simplicity. Also, the AIT bound is very general – it predicts a probability $P(f)$ that depends mainly on the function, and only weakly on the network. It may therefore not capture some variations in the bias that are due to details of the network architecture, and which may be important for practical applications.

For these reasons it is of interest to obtain a finer quantitative understanding of the simplicity bias of neural networks. Some work has been done in this direction, showing that infinitely wide neural networks are biased towards functions which are robust to changes in the input (De Palma et al., 2018), showing that "flatness" is connected to function smoothness (Wu et al., 2016), or arguing that low Fourier frequencies are learned first by a ReLU neural network (Rahaman et al., 2018; Yang & Salman, 2019). All of these papers take some notion of "smoothness" as tractable proxy for the complexity of a function. One generally expects smoother functions to be simpler, although this is clearly a very rough measure of the Kolmogorov complexity.

## 2  SUMMARY OF KEY RESULTS

In this paper we study how likely different Boolean functions, defined as $f : \{0,1\}^n \to \{0,1\}$, are obtained upon randomly chosen weights of neural networks. Our key results are aimed at fleshing out with more precision and rigour what the inductive biases of (very) simple neural networks are, and how they arise For this, we study the prior distribution over functions $P(f)$, upon random initialization of the parameters, which reflects the inductive bias of training algorithms that approximate Bayesian inference (see Appendix L for a detailed explanation, and data on how well SGD follows this behaviour). We focus our study on a notion of complexity, namely the "entropy," $H(f)$, of a Boolean function $f$, defined as the binary entropy of the fraction of possible inputs to $f$ that $f$ maps to $1$. This quantity essentially measures the amount of class imbalance of the function, and is complementary to previous works studying notions of smoothness as a proxy for complexity.

1. In Section 4 we study a simple perceptron with no threshold bias term, and with weights $w$ sampled from a distribution which is symmetric under reflections along the coordinate axes. Let the random variable $T$ correspond to the number of points in $\{0,1\}^n$ which that fall above the decision boundary of the network (i.e. T=$|\{x \in \{0,1\}^n : \langle w, x \rangle > 0\}|$) upon i.i.d. random initialisation of the weights. We prove that $T$ is distributed uniformly, i.e. $P(T = t) = 2^{-n}$ for $0 \le t < 2^n$. Let $\mathbb{F}_t$ be the set of all functions with $T = t$ that the perceptron can produce and let $|\mathbb{F}_t|$ be its size (cf. Definition 3.4). We expect $|\mathbb{F}_t|$ for $t \sim 2^{n-1}$ (high entropy) to be (much) larger than $|\mathbb{F}_t|$ for extreme values of $t$ (low entropy). The average probability of obtaining a particular function $f$ which maps $t$ inputs to $1$ is $2^{-n}/|\mathbb{F}_t|$. The perceptron therefore shows a strong bias towards functions with low entropy, in the sense that individual functions with low entropy have, on average, higher probability than individual functions with high entropy.

2. In Section 4.3, we show that within the sets $\mathbb{F}_t$, there is a further bias, and in some cases this is clearly towards simple functions which correlates with Lempel-Ziv complexity (Lempel & Ziv, 1976; Dingle et al., 2018), as predicted in (Valle-Pérez et al., 2018).

3. In Section 4.4, we show that adding a threshold bias term to a perceptron significantly increases the bias towards low entropy.

4. In Section 5.1, we provide a new expressivity bound for Boolean functions: DNNs with input size $n$, $l$ hidden layers each with width $n + 2^{n-1-\log_2 l} + 1$ and a single output neuron can express all $2^{2^n}$ Boolean functions over $n$ variables.

5. In Section 5.2 we generalise our results to neural networks with multiple layers, proving (in the infinite-width limit) that the bias towards low entropy increases with the number of ReLU-activated layers.

In Appendix J, we also show some empirical evidence that the results derived in this paper seem to generalize beyond the assumptions of our theoretical analysis, to more complicated data distributions (MNIST, CIFAR) and architectures (CNNs). Finally, in Appendix M, we show preliminary results on the effect of entropy-like biases in $P(f)$ on learning class-imbalanced data.

## 3 DEFINITIONS, TERMINOLOGY, AND NOTATION

**Definition 3.1** (DNNs). *Fully connected feed-forward neural networks with activations $\sigma$ and a single output neuron form a parameterised function family $f(x)$ on inputs $x \in \mathbb{R}^n$. This can be defined recursively, for $L$ hidden layers for $1 \leq l \leq L$, as*

$$f(x) = \mathbf{1}(h^{(L+1)}(x)),$$
$$h^{(l+1)}(x) = w_l \sigma(h^{(l)}) + b_l,$$
$$h^{(1)}(x) = w_0 x + b_0,$$

*where $\mathbf{1}(X)$ is the Heaviside step function defined as 1 if $X > 0$ and 0 otherwise, and $\sigma$ is an activation function that acts element-wise. The $w_l \in \mathbb{R}^{n_{l+l} \times n_l}$ are the weights, and $b_l \in \mathbb{R}^{n_{l+1}}$ are the threshold bias weights at layer $l$, where $n_l$ is the number of hidden neurons in the $l$-th layer. $n_{L+1}$ is the number of outputs (1 in this paper), and $n_0$ is the dimension of the inputs (which we will also refer to as $n$).*

We will refer to the whole set of parameters ($w_l$ and $b_l$, $1 \leq l \leq L$) as $\theta$. In the case of perceptrons we use $f_\theta(x) = \sigma(\langle w, x \rangle + b)$ to specify a network. We define the parameter-function map as in (Valle-Pérez et al., 2018) below.

**Definition 3.2** (Parameter-function map). *Consider a parameterised supervised model, and let the input space be $\mathbb{X}$ and the output space be $\mathbb{Y}$. The space of functions the model can express is $\mathcal{F} \subset \mathbb{Y}^{|\mathbb{X}|}$. If the model has $p$ real valued parameters, taking values within a set $\Theta \subseteq \mathbb{R}^p$, the parameter function map $\mathcal{M}$ is defined*

$$\mathcal{M} : \Theta \to \mathbb{F}$$
$$\theta \mapsto f_\theta$$

where $f_\theta$ is the function corresponding to parameters $\theta$.

In this paper we are interested in the Boolean functions that neural networks express. We consider the 0-1 Boolean hypercube $\{0,1\}^n$ as the input domain.

**Definition 3.3.** *The function $\mathcal{T}(f)$ is defined as the number of points in the hypercube $\{0,1\}^n$ that are mapped to 1 by the action of a neural network $f$.*

For example, for a perceptron this function is defined as,

$$\mathcal{T}(f) = \mathcal{T}(w, b) = \sum_{x \in \{0,1\}^n} \mathbf{1}(\langle x, w \rangle + b). \tag{2}$$

We will sometimes use $\mathcal{T}(w, b)$ if the neural network is a perceptron.

**Definition 3.4** ($\mathbb{F}_t$ and $P(t)$)**.** *We define the set $\mathbb{F}_t$ to be the set of functions expressible by some model $\mathcal{M}$ (e.g. a perceptron, a neural network) which all have the same value of $\mathcal{T}(f)$,*

$$\mathbb{F}_t = \{f \in \mathcal{F}_\mathcal{M} | \mathcal{T}(f) = t\}$$

*, where $\mathcal{F}_\mathcal{M}$ is the set of all functions expressible by $\mathcal{M}$. Given a probability measure $P$ on the weights $\theta$, we define the probability measure*

$$P(T = t) := P(\theta : f_\theta \in \mathbb{F}_t)$$

We can also define $\mathcal{T}(f)$ and $P(T = t)$ in the natural way for sets of input points other than $\{0, 1\}^n$, the context making clear what definition is being used.

**Definition 3.5.** *The entropy $H(f)$ of a Boolean function $f : \{0, 1\}^* \to \{0, 1\}$ is defined as $H(f) = -p \log_2 p - (1 - p) \log_2 (1 - p)$, where $p = \mathcal{T}_f / 2^n$. It is the binary entropy of the fraction $p$ of possible inputs to $f$ that $f$ maps to $1$ or equivalently, the binary entropy of the fraction of $1$'s in the right-hand column of the truth table of $f$.*

**Definition 3.6.** *We define the Boolean complexity $K_{\mathrm{Bool}}(f)$ of a function $f$ as the number of binary connectives in the shortest Boolean formula that expresses $f$.*

Note that Boolean complexity can be defined in other ways as well. For example, $K_{\mathrm{Bool}}(f)$ is sometimes defined as the number of connectives (rather than *binary* connectives) in the shortest formula that expresses $f$, or as the depth of the most shallow formula that expresses $f$. These definitions tend to give similar values for the complexity of a given function, and so they are largely interchangeable in most contexts. We use the definition above because it makes our calculations easier.

## 4 INTRINSIC BIAS IN A PERCEPTRON'S PARAMETER-FUNCTION MAP

In this section we study the parameter-function map of the perceptron (Rosenblatt, 1958), in many ways the simplest neural network. While it famously cannot express many Boolean functions – including XOR – it remains an important model system. Moreover, many DNN architectures include layers of perceptrons, so understanding this very basic architecture may provide important insight into the more complex neural networks used today.

### 4.1 ENTROPY BIAS IN A SIMPLE PERCEPTRON WITH $b = 0$ (NO THRESHOLD BIAS TERM)

Here we consider perceptrons $f_\theta(x) = \mathbf{1}(\langle w, x \rangle + b)$ without threshold bias terms, i.e. $b = 0$.

The following theorem shows that under certain conditions on the weight distribution, a perceptron with no threshold bias has a uniform $P(\theta : \mathcal{T}(f_\theta) = t)$. The class of weight distributions includes the commonly used isotropic multivariate Gaussian with zero mean, a uniform distribution on a centred cuboid, and many other distributions. The full proof of the theorem is in Appendix A.

**Theorem 4.1.** *For a perceptron $f_\theta$ with $b = 0$ and weights $w$ sampled from a distribution which is symmetric under reflections along the coordinate axes, the probability measure $P(\theta : \mathcal{T}(f_\theta) = t)$ is given by*

$$P(\theta : \mathcal{T}(f_\theta) = t) = \begin{cases} 2^{-n} & \text{if } 0 \leq t < 2^n \\ 0 & \text{otherwise} \end{cases}.$$

*Proof sketch.* We consider the sampling of the normal vector $w$ as a two-step process: we first sample the absolute values of the elements, giving us a vector $w_{\mathrm{pos}}$ with positive elements, and then we sample the signs of the elements. Our assumption on the probability distribution implies that each of the $2^n$ sign assignments is equally probable, each happening with a probability $2^{-n}$. The key of the proof is to show that for any $w_{\mathrm{pos}}$, each of the sign assignments gives a distinct value of $T$ (and because there are $2^n$ possible sign assignments, for any value of $T$, there is exactly one sign assignment resulting in a normal vector with that value of $T$). This implies that, provided all sign assignments of any $w_{\mathrm{pos}}$ are equally likely, the distribution on $T$ is uniform. □

A consequence of Theorem 4.1 is that the average probability of the perceptron producing a particular function $f$ with $\mathcal{T}(f) = t$ is given by

$$\langle P(f) \rangle_t = \frac{2^{-n}}{|\mathbb{F}_t|}, \tag{3}$$

where $\mathbb{F}_t$ denotes the set of Boolean functions that the perceptron can express which satisfy $\mathcal{T}(f) = t$, and $\langle \cdot \rangle_t$ denotes the average (under uniform measure) over all functions $f \in \mathbb{F}_t$.

We expect $|\mathbb{F}_t|$ to be much smaller for more extreme values of $t$, as there are fewer distinct possible functions with extreme values of $t$. This would imply a bias towards low *entropy* functions. By way of an example, $|\mathbb{F}_0| = 1$ and $|\mathbb{F}_1| = n$ (since the only Boolean functions $f$ a perceptron can express which satisfy $\mathcal{T}(f) = 1$ have $f(x) = 1$ for a single one-hot $x \in \{0,1\}^n$), implying that $\langle P(f) \rangle_0 = 2^{-n}$ and $\langle P(f) \rangle_1 = 2^{-n}/n$.

Nevertheless, the probability of functions within a set $\mathbb{F}_t$ is unlikely to be uniform. We find that, in contrast to the overall entropy bias, which is independent of the shape of the distribution (as long as it satisfies the right symmetry conditions), the probability $P(f)$ of obtaining function $f$ within a set $\mathbb{F}_t$ can depend on distribution shape. Nevertheless, for a given distribution shape, the probabilities $P(f)$ are independent of scale of the shape, e.g. they are independent of the variance of the Gaussian, or the width of the uniform distribution. This is because the function is invariant under scaling all weights by the same factor (true only in the case of no threshold bias). We will address the probabilities of functions within a given $\mathbb{F}_t$ further in Section 4.3.

## 4.2 Simplicity bias of the $b = 0$ perceptron

The entropy bias of Theorem 4.1 entails an overall bias towards low Boolean complexity. In Theorem B.1 in Appendix B we show that the Boolean complexity of a function $f$ is bounded by[1]

$$K_{\text{Bool}}(f) < 2 \times n \times \min(\mathcal{T}(f), 2^n - \mathcal{T}(f)). \tag{4}$$

Using Theorem 4.1 and Equation (4), we have that the probability that a randomly initialised perceptron expresses a function $f$ of Boolean complexity $k$ or greater is upper bounded by

$$P(K_{\text{Bool}}(f) \geq k) < 1 - \frac{k \times 2^{-n} \times 2}{2 \times n} = 1 - \frac{k}{2^n \times n}. \tag{5}$$

Uniformly sampling functions would result in $P(K_{\text{Bool}}(f) \geq k) \approx 1 - 2^{k-2^n}$ which for intermediate $k$ is much larger than Equation (5). Thus from entropy bias alone, we see that the perceptron is much more likely to produce simple functions than complex functions: it has an inductive bias towards simplicity. This derivation is complementary to the AIT arguments from *simplicity bias* (Dingle et al., 2018; Valle-Pérez et al., 2018), and has the advantage that it also proves that bias exists, whereas AIT-based simplicity bias arguments presuppose bias.

To empirically study the inductive bias of the perceptron with $b = 0$, we sampled over many random initialisations with weights drawn from Gaussian or uniform distributions and input size $n = 7$. As can be seen in Figure 1a and Figure 1b, the probability $P(f)$ that function $f$ obtains varies over many orders of magnitude. Moreover, there is a clear simplicity bias upper bound on this probability, which, as predicted by Eq. 1, decreases with increasing Lempel-Ziv complexity ($K_{LZ}(f)$) (using a version from (Dingle et al., 2018) applied to the Boolean functions represented as strings of bits, see Appendix E). Similar behaviour was observed in (Valle-Pérez et al., 2018) for a FCN network. Moreover it was also shown there that Lempel-Ziv complexity for these Boolean functions correlates with approximations to the Boolean complexity $K_{\text{Bool}}$. A one-layer neural network ( Figure 1c) shows stronger bias than the perceptron, which may be expected because the former has a much larger expressivity. A rough estimate of the slope $a$ in Eq. 1 from (Dingle et al., 2018) suggests that $a \sim \log_2(N_O)/\max_{f \in \mathbb{O}}(\tilde{K}(f))$ where $\mathbb{O}$ is the set of all Boolean functions the model can produce, and $N_O$ is the number of such functions. The maximum $K(f)$ may not differ that much between the one layer network and the perceptron, but $N_O$ will be much larger in former than in the latter.

In Appendix D we also show rank plots for the networks from Figure 1. Interestingly, at larger rank, they all show a Zipf like power-law decay, which can be used to estimate $N_O$, the total number of

---

[1]A tighter bound is given in Theorem B.2, but this bound lacks any obvious closed form expression.

Boolean functions the network can express. We also note that the rank plots for the perceptron with $b = 0$ with Gaussian or uniform distributions of weights are nearly indistinguishable, which may be because the overall rank plot is being mainly determined by the entropy bias.

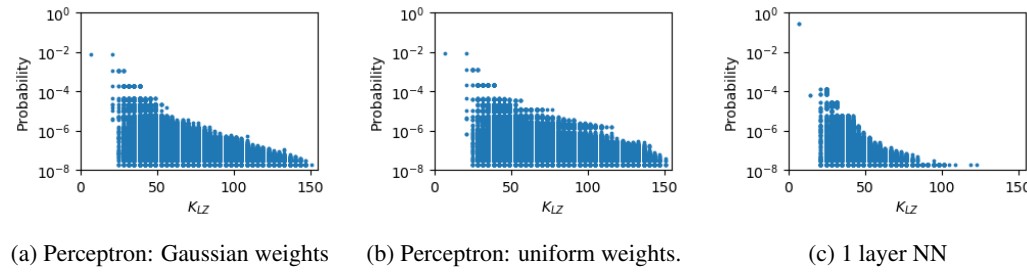

(a) Perceptron: Gaussian weights    (b) Perceptron: uniform weights.    (c) 1 layer NN

Figure 1: Probability $P(f)$ that a function obtains upon random choice of parameters versus Lempel Ziv complexity $K_{LZ}(f)$ for (a) an $n = 7$ perceptron with $b = 0$ and weights sampled from a Gaussian distributions, (b) an $n = 7$ perceptron with $b = 0$ and weights sampled from a uniform distribution centred at 0 and (c) a 1-hidden layer neural network (with 64 neurons in the hidden layer). Weights $w$ and the threshold bias terms are sampled from $\mathcal{N}(0, 1)$. For all cases $10^8$ samples were taken and frequencies less than 2 were eliminated to reduce finite sampling effects. We present the graphs with the same scale for ease of comparison.

## 4.3   BIAS WITHIN $\mathbb{F}_t$

In Figure 2 we compare a rank plot for all functions expressed by an $n = 7$ perceptron with $b = 0$ to the rank plots for functions with $\mathcal{T}(f) = 47$ and $\mathcal{T}(f) = 64$. To define the rank, we order the functions by decreasing probability, and then the rank of a function $f$ is the index of $f$ under this ordering (so the most probable function has rank 1, the second rank 2 and so on). The highest probability functions in $\mathbb{F}_{64}$ have higher probability than the highest in $\mathbb{F}_{47}$ because the former allows for simpler functions (such as 0101..), but for both sets, the maximum probability is still considerably lower than the maximum probability functions overall.

In Appendix E we present further empirical data that suggests that these probabilities are bounded above by Lempel-Ziv complexity (in agreement with (Valle-Pérez et al., 2018)). However, in contrast to Theorem 4.1 which is independent of the parameter distribution (as long as they are symmetric), the distributions within $\mathbb{F}_t$ are different for the Gaussian and uniform parameter distributions, with the latter showing less simplicity bias within a class of fixed $t$ (see Appendix E.1).

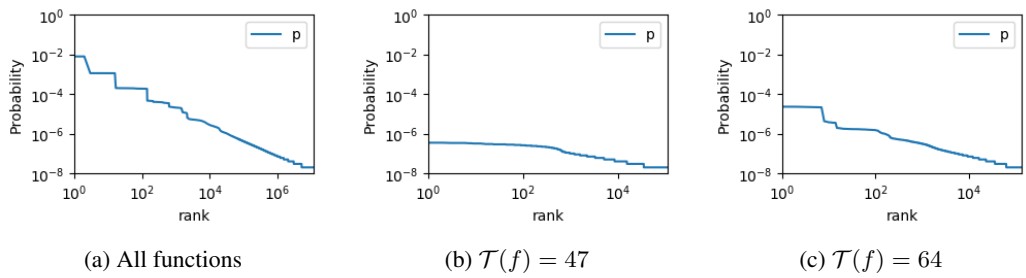

(a) All functions    (b) $\mathcal{T}(f) = 47$    (c) $\mathcal{T}(f) = 64$

Figure 2: Probability $P(f)$ vs rank for functions for a perceptron with $n = 7$, $\sigma_b = 0$, and weights sampled from independent Gaussian distributions. In Figures 2b and 2c the functions are ranked within their respective $\mathbb{F}_t$. The seven highest probability functions in Figure 2c are $f = 0101\ldots$ and equivalent functions obtained by permuting the input dimensions – note that these are very simple functions (simpler than the simplest functions that satisfy $\mathcal{T}(f) = 47$).

In Appendix F, we give further arguments for simplicity bias, based on the set of constraints that needs to be satisfied to specify a function. Every function $f$ can be specified by a minimal set of linear conditions on the weight vector of the perceptron, which correspond to the boundaries of the cone in weight space producing $f$. The Kolmogorov complexity of conditions should be close to that of the functions they produce as they are related to the functions in a one-to-one fashion, via a simple procedure. In Appendix F, we focus on conditions which involve more than two weights, and show that within each set $\mathbb{F}_t$ there exists one function with as few as 1 such conditions, and that there exists a function with as many as $n-2$ such conditions. We also compute the set of necessary conditions (up to permutations of the axes) explicitly for functions with small $t$, and find that the range in the number and complexity of the conditions appears to grow with $t$, in agreement, with what we observe in Figure 2 for the range of complexities. More generally, we find that complex functions typically need more conditions than simple functions do. Intuitively, the more conditions needed to specify a function, the smaller the volume of parameters that can generate the function, so the lower its *a-priori* probability.

### 4.4 EFFECT OF $b$ (THE THRESHOLD BIAS TERM) ON $P(t)$

We next study the behaviour of the perceptron when we include the threshold bias term $b$, sampled from $\mathcal{N}(0, \sigma_b)$, while still initialising the weights from $\mathcal{N}(0, 1)$, as in Section 4.1. We present results for $n = 7$ in Figure 3. Interestingly, for infinitesimal $\sigma_b$, $P(T = 0)$ is less than for $b = 0$ (See Appendix C), but then for increasing $\sigma_b$ it rapidly grows larger than $1/2^n$ and in the limit of large $\sigma_b$ asymptotes to $1/2$ (see Figure 3b). It's not hard to see where this asymptotic behaviour comes from, a large positive or negative $b$ means all inputs are mapped to true (1) or false (0) respectively.

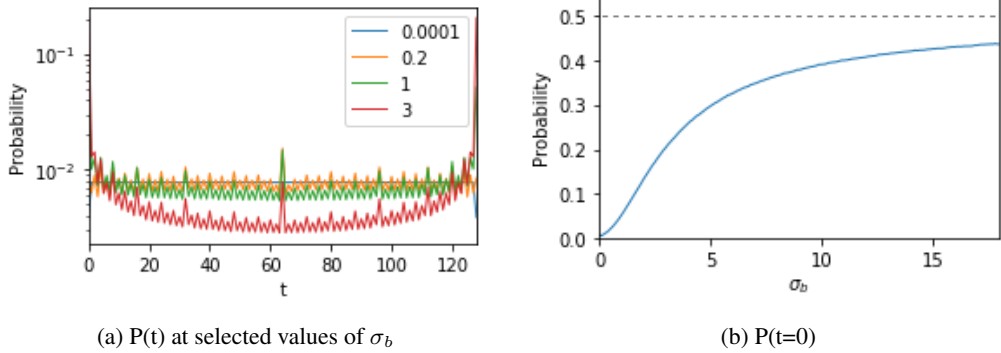

(a) P(t) at selected values of $\sigma_b$          (b) P(t=0)

Figure 3: Effect of adding a bias term sampled from $\mathcal{N}(0, \sigma_b)$ to a perceptron with weights sampled from $\mathcal{N}(0, 1)$. (a) Increasing $\sigma_b$ increases the bias against entropy, and with a particular strong bias towards $t = 0$ and $t = 2^n$. (b) $P(t = 0)$ increases with $\sigma_b$ and asymptotes to $1/2$ in the limit $\sigma_b \to \infty$.

## 5 ENTROPY BIAS IN MULTI-LAYER NEURAL NETWORKS

We next extend results from Section 4 to multi-layer neural networks, with the aim to comment on the behaviour of $P(T = t)$ as we add hidden layers with ReLU activations.

To study the bias in the parameter-function map of neural networks, it is important to first understand the expressivity of the networks. In Section 5.1, we produce a (loose) upper bound on the minimum size of a network with ReLU activations and $l$ layers that is maximally expressive over Boolean functions. We comment on how sufficiently large expressivity implies a larger bias towards low entropy for models with similarly shaped distribution over $T$ (when compared to the perceptron).

In Section 5.2, we prove, in the limit of infinite width, that adding ReLU activated layers causes the moments of $P(T = t)$ to increase, . This entails a lower expected entropy for neural networks with more hidden layers. We empirically observe that the distribution of $T$ becomes convex (with input $\{0, 1\}^n$) with the addition of ReLU activated layers for neural networks with finite width.

## 5.1 Expressivity conditions for DNNs

We provide upper bounds on the minimum size of a DNNs that can model all Boolean functions. We use the notation $\langle n_0, n_1, \ldots, n_L, n_{L+1} \rangle$ to denote a neural network with ReLU activations and of the form given in Definition 3.1.

**Lemma 5.1.** *A neural network with layer sizes $\langle n, 2^{n-1}, 1 \rangle$, threshold bias terms, and ReLU activations can express all Boolean functions over $n$ variables (also found in (Raj, 2018)). See Appendix G for proof.*

**Lemma 5.2.** *A neural network with $l$ hidden layers, layer sizes $\langle n, (n + 2^{n-1}/l + 1), \ldots, (n + 2^{n-1}/l + 1), 1 \rangle$, threshold bias terms, and ReLU activations can express all Boolean functions over $n$ variables. See Appendix G for proof.*

Note that neither of these bounds are (known to be) tight. Lemma 5.1 says that a network with one hidden layer of size $2^{n-1}$ can express all Boolean functions over $n$ variables. We know that a perceptron with $n$ input neurons (and a threshold bias term) can express at most $2^{n^2}$ Boolean functions ((Anthony, 2001), Theorem 4.3), which is significantly less than the total number of Boolean functions over $n$ variables, which is $2^{2^n}$. Hence there is a very large number of Boolean functions that the network with a (sufficiently wide) hidden layer can express, but the perceptron cannot. The vast majority of these functions have high entropy (as almost all Boolean functions do). Moreover, we observe that the measure $P(T = t)$ is convex in the case of the more expressive neural networks, as discussed in section Section 5.2. This suggests that the networks with hidden layers have a much stronger relative bias towards low entropy functions than the perceptron does, which is also consistent with the stronger simplicity bias found in Figure 1.

We further observe from Lemma 5.2 that the number of neurons can be kept constant and spread over multiple layers without loss of expressivity for a Boolean classifier (provided the neurons are evenly spread across the layers).

## 5.2 How multiple layers affect the bias

We next consider the effect of addition of ReLU activated layers on the distribution $P(t)$. Of course adding even just one layer hugely increases expressivity over a perceptron. Therefore, even if the distribution of $P(t)$ would not change, the average probability of functions in a given $\mathbb{F}_t$ could drop significantly due to the increase in expressivity.

However, we observe that for inputs $\{0, 1\}^n$, $P(t)$ becomes more convex when more ReLU-activated hidden layers are added, see Figure 4. The distribution appears to be monotone on either side of $t = 2^{n-1}$ and relatively flat in the middle, even with the addition of 8 intermediate layers[2]. In particular, we show in Figure 4 that for large number of layers, or large $\sigma_b$, the probabilities for $P(t = 0)$ (and by symmetry, in the infinite width limit, also $P(t = 2^n)$) each asymptotically reach $\frac{1}{2}$, and thus take up the vast majority of the probability weight.

We now prove some properties of the distribution $P(t)$ for DNNs with several layers.

**Lemma 5.3.** *The probability distribution on T for inputs in $\{0, 1\}^n$ of a neural network with linear activations and i.i.d. initialisation of the weights is independent of the number of layers and the layer widths, and is equal to the distribution of a perceptron. See Appendix H for proof.*

While it is trivial that such a linear network has the same expressivity as a perceptron, it may not be obvious that the entropy bias is identical.

**Lemma 5.4.** *Applying a ReLU function in between each layer produces a lower bound on $P(T = 0)$ such that $P(T = 0) \geq 2^{-n}$. See Appendix H for proof.*

This lemma shows that a DNN with ReLU functions is no less biased towards the lowest entropy function than a perceptron is. We prove a more general result in the following theorem which concerns the behaviour of the average entropy $\langle H(t) \rangle$ (where the average upon random sampling of parameters) as the number of layers grows. The theorem shows that the bias towards low entropy becomes stronger as we increase the number of layers, for any distribution of inputs. We rely

---

[2]Note that this is when the input is $\{0, 1\}^n$. This is not true for all input distributions. See, e.g. Figure 4d. The change in probability for other distributions can be more complex.

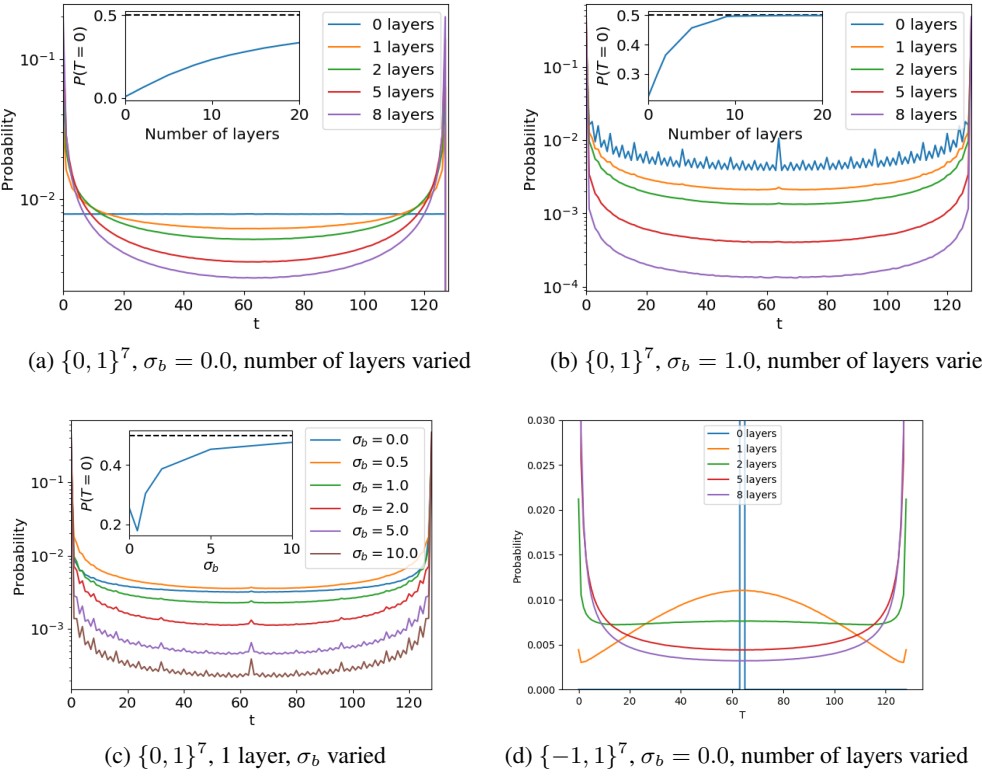

Figure 4: **$P(T = t)$ becomes on average more biased towards low entropy for increasing number of layers or increasing $\sigma_b$.** Here we use $n = 7$ input layers, with input $\{0, 1\}^7$ (*centered data*) or $\{-1, 1\}^7$ (*uncentered data*) The hidden layers are of width $2^{n-1} = 64$ to guarantee full expressivity. $\sigma_w = 1.0$ in all cases. The insets show how $P(t = 0)$ asymptotes to $\frac{1}{2}$ with increasing layers or $\sigma_b$.

on previous work that shows that in the infinite width limit, neural networks approach a Gaussian process (Lee et al. (2018); Garriga-Alonso et al. (2018); Novak et al. (2018); Matthews et al. (2018); Yang (2019)), which for the case of fully-connected ReLU networks, has an analytic form (Lee et al., 2018).

**Theorem 5.5.** *Let $\mathbb{S}$ be a set of $m = |\mathbb{S}|$ input points in $\mathbb{R}^n$. Consider neural networks with i.i.d. Gaussian weights with variances $\sigma_w^2/\sqrt{n}$ and biases with variance $\sigma_b$, in the limit where the width of all hidden layers $n$ goes to infinity. Let $N_1$ and $N_2$ be such neural networks with $L$ and $L + 1$ infinitely wide hidden layers, respectively, and no bias. Then, the following holds: $\langle H(T) \rangle$ is smaller than or equal for $N_2$ than for $N_1$. It is strictly smaller if there exist pairs of points in $\mathbb{S}$ with correlations less than 1. If the networks have sufficiently large threshold bias ($\sigma_b > 1$ is a sufficient condition), the result above also holds. For smaller bias, the result holds only for a sufficiently large number of layers.*

See Appendix H for a proof of Theorem 5.5. Theorem 5.5 is complementary to Theorem 4.1, in that the former only proves that the bias towards low entropy increases with depth, and the later proves conditions on the data that guarantee bias toward low entropy on the "base case" of 0 layers. We show in Figure 4 that when $\sigma_b = 0$, the bias towards low entropy indeed becomes monotonically stronger as we increase the number of ReLU layers, for both inputs in $\{0, 1\}^n$ as well as for centered data $\{-1, 1\}^n$.

For centered inputs $\{-1, 1\}^n$, the perceptron with $b = 0$ shows rather unusual behaviour. The distribution is completely peaked around $t = 2^{n-1}$ because every input mapping to 1 has the opposite input mapping to 0. Not surprisingly, its expressivity is much lower than the equivalent perceptron with $\{0, 1\}^n$ (as can be seen in Figure 6a in Appendix D). Nevertheless, in Figure 4d we see that

as the number of layers increases, the behaviour rapidly resembles that of uncentered data (In Appendix K we also show that the bias toward low entropy is also recovered as we increase $\sigma_b$). So far this is the only exception we have found to the general bias to low entropy we observe for all other systems (see also Appendix J). We therefore argue that this is a singular result brought about by particular symmetries of the perceptron with zero bias term. The fact that there is an exception does not negate our general result which we find holds much more generally.

The insets of in Figure 4 show that the two trivial functions asymptotically dominate in the limit of large numbers of layers. We note that recent work ((Lee et al., 2018; Luther & Seung, 2019)) has also pointed out that for fully-connected ReLU networks in the infinite-width infinite-depth limit, all inputs become asymptotically correlated, so that the networks will tend to compute the constant function. Here we give a quantitative characterisation of this phenomenon for any number of layers.

Some interesting recent work (Yang & Salman, 2019) has shown that certain choices of hyperparameters lead to networks which are a priori unbiased, that is the $P(f)$ appears to be uniform. In Appendix I we show that this result is due to a choice of hyperparameters that lie deep in the chaotic region defined in (Poole et al., 2016). The effect therefore depends on the choice of activation function (it can occur for say tanh and erf, but most likely not ReLU), and we are studying it further.

# 6 DISCUSSION AND FUTURE WORK

In Section 4 Theorem 4.1, we have proven the existence of an intrinsic bias towards Boolean functions of low entropy in a perceptron with no threshold bias term, such that $P(T = t) = 2^{-n}$ for $0 \leq t < 2^n$. This result puts an upper bound on the probability that a perceptron with no threshold bias term will be initialised to a Boolean function with at least a certain Boolean complexity. Adding a threshold term in general increases the bias towards low entropy.

We also study how the entropy bias is affected by adding a threshold bias term or ReLU-activated hidden layers. One of our main results, Theorem 5.5, proves that adding layers to a feed-forward neural network with ReLU activations makes the bias towards low entropy stronger. We also show empirically that the bias towards low entropy functions is further increased when a threshold bias term with high enough variance is added. Recently, (Luther & Seung, 2019) have argued that batch normalisation (Ioffe & Szegedy, 2015) makes ReLU networks less likely to compute the constant function (which has also been experimentally shown in (Page, 2019)). If batch norm increases the probability of high entropy functions, it could help explain why batch norm improves generalisation for (typically class balanced) datasets. We leave further exploration of the effect of batch normalisation on a-priori bias to future work.

Simplicity bias within the set of constant $t$ functions $\mathbb{F}_t$ is affected by the choice of initialisation, even when the entropy bias is unaffected. This indicates that there are further properties of the parameter-function map that lead to a simplicity bias. In Section 4.3, we suggest that the complexity of the conditions on $w$ producing a function should correlate with the complexity of the function, and we conjecture that more complex conditions correlate with a lower probability.

We note that the *a priori* inductive bias we study here is for a randomly initialised network. If a network is trained on data, then the optimisation procedure (for example SGD) may introduce further biases. In Appendix L, we give some evidence that the bias at initialization is the main driver of the inductive bias on SGD-trained networks. Furthermore, in Appendix M, we show preliminary results on how bias in $P(f)$ can affect learning in class-imbalanced problems. This suggests that understanding properties of $P(f)$ (like those we study in this paper), can help design architectures with desired inductive biases.

Simplicity bias in neural networks (Valle-Pérez et al., 2018) offers an explanation of why DNNs work in the highly overparameterised regime. DNNs can express an unimaginably large number of functions that will fit the training data, but almost all of these will give extremely poor generalisation. Simplicity bias, however, means that a DNN will preferentially choose low complexity functions, which should give better generalisation. Here we have shown some examples where changing hyperparameters can affect the bias further. This raises the possibility of explicitly designing biases to optimise a DNN for a particular problem.

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

# A    PROOF OF UNIFORMITY

For convenience we repeat some notation we use in this section. Let $\{0,1\}^n$ be the set of vertices of the $n$-dimensional hypercube. We use $\langle \cdot, \cdot \rangle$ to refer to the standard inner product in $\mathbb{R}^n$. Define the function $\mathcal{T} : \mathbb{R}^n \to \mathbb{N}$ as the number of vertices of the hypercube that are above the hyperplane with normal vector $w$ and that passes through the origin. Formally $\mathcal{T}(w) = |x \in \{0,1\}^n : \langle w, x \rangle > 0|$. We use $\odot$ for element-wise multiplication of two vectors.

We slightly abuse notation and denote the probability density function corresponding to a probability measure $P$, with the same symbol, $P$. The arguments of the function or context should make clear which one is meant.

**Proof strategy.** We consider the sampling of the normal vector $w$ as a two-step process: we first sample the absolute values of the elements, giving us a vector $w_{\text{pos}}$ with positive elements[3], and then we sample the signs of the elements. Our assumption on the probability distribution implies that each of the $2^n$ sign assignments is equally probable, each happening with a probability $2^{-n}$. The key of the proof is to show that for any $w_{\text{pos}}$, each of the sign assignments gives a distinct value of $T$ (and because there are $2^n$ possible sign assignments, for any value of $T$, there is exactly one sign assignment resulting in a normal vector with that value of $T$). This implies that, provided all sign assignments of any $w_{\text{pos}}$ are equally likely, the distribution on $T$ is uniform.

**Theorem 4.1** *Let $P$ be a probability measure on $\mathbb{R}^n$, which is symmetric under reflections along the coordinate axes, so that $P(x) = P(Rx)$, where $R$ is a reflection matrix (a diagonal matrix with elements in $\{-1, 1\}$). Let the weights of a perceptron without bias, $w$, be distributed according to $P$. Then $P(T = t)$ is the uniform measure.*

Before proving the theorem, we first need a definition and a lemma.

**Definition** We define the function mapping a vector from $\{-1, 1\}^n$ (which we interpret as the signature of the weight vector), and a vector of nonnegative reals (which we interpret as the absolute values of the elements of the weight vector) to the value of $t$ of the corresponding weight vector:

$$K : \{-1, 1\}^n \times \mathbb{R}^n_{\geq 0} \to \{0, 1, ..., 2^n - 1\}$$
$$(\sigma, a) \mapsto \mathcal{T}(\sigma \odot a) \tag{6}$$

**Lemma A.1.** *The function $K$ is bijective with respect to its first argument, for any value of its second argument except for a set of measure $0$.*

*Proof of Lemma A.1.* Because the cardinality of the codomain of $K$ is the same as the domain of its first argument, it is enough to prove injectivity of $K$ with respect to its first argument.

Fix $a \in \mathbb{R}^n_{\geq 0}$ satisfying that the following set has cardinality $3^n$:

$$\tilde{\mathbb{S}}_a = \{\langle x, a \rangle : x \in \{-1, 0, 1\}^n\}.$$

Note that the set of $a$ in which some pair of elements in the definition of $\mathbb{S}_a$ is equal has measure zero, because their equality implies that $a$ lies within a hyperplane in $\mathbb{R}^n$. Let us also define the set of subsums of elements of $a$:

$$\mathbb{S}_a = \{\langle x, a \rangle : x \in \{0, 1\}^n\}.$$

which has cardinality $2^n$ for the $a$ considered.

Now, consider a natural bijection $J_a : \{-1, 1\}^n \to \mathbb{S}_a$ induced by the bijective mapping of signatures $\sigma \in \{-1, 1\}^n$ to vertices of the hypercube $\{0, 1\}^n$ by mapping $-1$ to $0$ and $1$ to $1$. To be more precise, $J_a(\sigma) = \sum_{i=1}^n a_i \frac{(\sigma(i)+1)}{2}$.

Then, we claim that

$$K(\sigma, a) = |\{s \in \mathbb{S}_a : s < J_a(\sigma)\}| \tag{7}$$

This implies that for the $a$ we have fixed $K$ is injective, for if two $\sigma$ mapped to the same value, their corresponding value of $J_a(\sigma)$ should be the same, giving a contradiction. So it only remains to prove equation 7.

---

[3]almost surely, assuming 0 has zero probability measure

Let us also first denote, for $x \in \{0,1\}^n$ and $s, s' \in \mathbb{S}_a$

$$\Sigma(x) := \langle x, a \rangle$$
$$s \cap s' := \left\langle (\Sigma^{-1}(s) \cap \Sigma^{-1}(s')), a \right\rangle$$
$$s \cup s' := \left\langle (\Sigma^{-1}(s) \cup \Sigma^{-1}(s')), a \right\rangle$$
$$\bar{s} := \left\langle (\overline{\Sigma^{-1}(s)}), a \right\rangle$$

where we interpret elements of $\{0,1\}^n$ as subsets of $\{1, ..., n\}$. The notation above lets us interpret subsums in $S_a$ as subsets of entries of $a$. Note that $\Sigma^{-1}$ is well defined for the fixed $a$ we are considering.

Now, let $\sigma \in \{-1,1\}^n$, $s' = J_a(\sigma)$, and consider an $s \in \mathbb{S}_a$ such that $s < s'$. Then $s \cap s' + s \cap \bar{s}' = s < s'$ so

$$- s \cap s' + s' - s \cap \bar{s}' > 0 \tag{8}$$

Now, let the operation $^*$ (we omit dependence on $s'$) be defined for any $u \in \mathbb{S}_a$ as $u^* := u \cap \bar{s}' + (s' - u \cap s') \in \mathbb{S}_a$. Since $s' = J_a(\sigma)$ we have

$$\left\langle (\sigma \odot a), \Sigma^{-1}(s^*) \right\rangle = -s \cap \bar{s}' + (s' - s \cap s').$$

Using equation 8,

$$\left\langle (\sigma \odot a), \Sigma^{-1}(s^*) \right\rangle > 0 \iff s < s'.$$

Therefore, all the points $\Sigma^{-1}(s^*)$ for $s < s'$ are above the hyperplane with normal $(\sigma \odot a)$, and all points $\Sigma^{-1}(s^*)$ for $s \geq s'$ are below or precisely on the hyperplane. All that is left is to show the converse, all points which are above the hyperplane are $\Sigma^{-1}(s^*)$ for one and only one $s < s'$. It suffices to show that the operation $^*$ is injective for all $s$ (as bijectivity follows from the domain and codomain being the same). By contradiction, let $s$ and $u$ map to the same value under $^*$, then $s \cap s' - s \cap \bar{s}' = u \cap s' - u \cap \bar{s}'$, which implies $s \cap s' = u \cap s'$ and $s \cap \bar{s}' = u \cap \bar{s}'$, for the $a$ we are considering, and so $s = u$. Therefore $^*$ is injective, and equation 7 follows. $\square$

*Proof of Theorem 4.1.*

$$P(T = t') = P(w : \mathcal{T}(w) = t') = \int \mathbf{1}_{\mathcal{T}(w)=t'} P(w) d^n w$$

Now, we can divide the integral into the quadrants corresponding to different signatures of $w$, and we can let $P(w) = \frac{1}{2^n} \tilde{P}(|w|)$, because it is symmetric under reflections of the coordinate axes.

$$P(w : \mathcal{T}(w) = t') = \sum_{\sigma \in \{-1,1\}^n} \int_{\mathbb{R}^n_{\geq 0}} \frac{1}{2^n} \tilde{P}(a) d^n a \mathbf{1}_{\mathcal{T}(\sigma \odot a)=t'}$$

$$= \frac{1}{2^n} \int_{\mathbb{R}^n_{\geq 0}} \tilde{P}(a) d^n a \sum_{\sigma \in \{-1,1\}^n} \mathbf{1}_{\mathcal{T}(\sigma \odot a)=t'}$$

$$= \frac{1}{2^n} \int_{\mathbb{R}^n_{\geq 0}} \tilde{P}(a) d^n a \cdot 1$$

$$= \frac{1}{2^n}.$$

The third equality follows from Lemma A.1. Indeed, bijectivity implies that for any $a$, except for a set of measure 0, there is one and only one signature which results in $t'$.

$\square$

# B   BOUNDING BOOLEAN FUNCTION COMPLEXITY, $K_{Bool}$, WITH $t$

**Theorem B.1.** $n \times \min(t, 2^n - t) - 1$ *is an upper bound on the complexity of Boolean functions $f$ for which $\mathcal{T}(f) = t$.*

*Proof.* Let $f : \{0, 1\}^n \to \{0, 1\}$ be a function s.t. $\mathcal{T}(f) = t$.

Let $x_1 \ldots x_n$ be propositional variables and let each assignment to $x_1 \ldots x_n$ correspond to a vector in $\{0, 1\}^n$ in the straightforward way.

Let $\phi$ be the Boolean formula $\bigvee\{\bigwedge\{$if $v_i = 1$ then $x_i$ else $\neg x_i \mid v_i \in v\} \mid v \in \{0, 1\}^n, f(v) = 1\}$. The formula $\phi$ expresses $f$ as a Boolean formula in Disjunctive Normal Form (DNF).

Let $\psi$ be the Boolean formula $\bigwedge\{\bigvee\{$if $v_i = 0$ then $x_i$ else $\neg x_i \mid v_i \in v\} \mid v \in \{0, 1\}^n, f(v) = 0\}$. The formula $\psi$ expresses $f$ as a Boolean formula in Conjunctive Normal Form (CNF).

Since $f$ maps $t$ out of the $2^n$ vectors in $\{0, 1\}^n$ to 1 it must be the case that $\phi$ has $t$ clauses and $\psi$ has $2^n - t$ clauses. Each clause contains $n - 1$ binary connectives, and there is one connective between each clause. Hence $\phi$ contains $n \times t - 1$ binary connectives and $\psi$ contains $n \times (2^n - t) - 1$ binary connectives. Therefore $f$ is expressed by some Boolean formula of complexity $n \times \min(t, 2^n - t) - 1$.

Since $f$ was chosen arbitrarily, if a function $f : \{0, 1\}^n \to \{0, 1\}$ maps $t$ inputs to 1 then the complexity of $f$ is at most $n \times \min(t, 2^n - t) - 1$. $\qquad\square$

**Theorem B.2.** *Let $C$ be a defined recursively as follows;*
$C(n, 0) = 0$
$C(n, 2^n) = 0$
$C(n, 1) = n - 1$
$C(n, 2^n - 1) = n - 1$
$C(n, t) = C(n - 1, \lceil t/2 \rceil) + C(n - 1, \lfloor t/2 \rfloor) + 2$

*Then $C(n, t)$ is an upper bound on the complexity of Boolean functions $f$ for which $\mathcal{T}(f) = t$.*

*Proof.* Let P($n$) be that $C(n, t)$ is an upper bound on the complexity of Boolean functions $f$ over $n$ variables s.t. $\mathcal{T}(f) = t$.

Base case P(1): If $\phi$ is a Boolean formula defined over 1 variable then $\phi$ is equivalent to True, False, $x_1$, or $\neg x_1$. We can see by exhaustive enumeration that P(1) holds in each of these four cases.

Inductive step P($n$) $\to$ P($n + 1$): Let $\phi$ be a Boolean formula defined over $n + 1$ variables s.t. $\mathcal{T}(\phi) = t$.

**Case 1.** $t = 0$: If $t = 0$ then $\phi \equiv$ False, and so the complexity of $\phi$ is 0. Hence the inductive step holds.
**Case 2.** $t = 2^n$: If $t = 2^n$ then $\phi \equiv$ True, and so the complexity of $\phi$ is 0. Hence the inductive step holds.
**Case 3.** $t = 1$: If $t = 1$ then $\phi$ has just a single satisfying assignment. If this is the case then $\phi$ can be expressed as a formula of length $n$ written in Disjunctive Normal Form, and hence the inductive step holds.
**Case 4.** $t = 2^n - 1$: If $t = 2^n - 1$ then $\phi$ has just a single non-satisfying assignment. If this is the case then $\phi$ can be expressed as a formula of length $n$ written in Conjunctive Normal Form, and hence the inductive step holds.
**Case 5.** $1 < t$ and $t < 2^n - 1$: If $\phi$ is a Boolean formula defined over $n + 1$ variables then $\phi$ is logically equivalent to a formula $(x_{n+1} \wedge \psi_1) \vee (\neg x_{n+1} \wedge \psi_2)$, where $\psi_1$ and $\psi_2$ are defined over $x_1 \ldots x_n$. Let $t_1$ be the number of assignments to $x_1 \ldots x_n$ that are mapped to 1 by $\psi_1$, and let $t_2$ be the corresponding value for $\psi_2$.

By the inductive assumption the complexity of $\psi_1$ and $\psi_2$ is bounded by $C(n, t_1)$ and $C(n, t_2)$ respectively. Therefore, since $\phi \equiv (x_{n+1} \wedge \psi_1) \vee (\neg x_{n+1} \wedge \psi_2)$ it follows that the complexity of $\phi$ is at most $C(n, t_1) + C(n, t_2) + 2$. Since $t_1 + t_2 = t$, and since $C(n, t_a) < C(n, t_b)$ if $t_b$ is closer to $2^{n-1}$ than $t_a$ is (lemma B.3), it follows that the complexity of $\phi$ is bounded by $C(n, t) = C(n - 1, \lceil t/2 \rceil) + C(n - 1, \lfloor t/2 \rfloor) + 2$.

Since Case 1 to 5 are exhaustive the inductive step holds. $\qquad\square$

**Lemma B.3.** *If $t + 1 \le 2^{n-1}$ then $C(n, t) < C(n, t + 1)$.*

*Proof.* Let P($n$) be that if $t + 1 \le 2^{n-1}$ then $C(n, t) < C(n, t + 1)$.

Base case P(2): We can see that $C(4,0) = 1$, $C(4,1) = 2$, $C(4,2) = 4$, $C(4,3) = 2$ and $C(4,4) = 1$. By exhaustive enumeration we can see that P(2) holds.

Inductive step P(n) → P(n + 1):

**Case 1.** *t is even*:

$$
\begin{aligned}
C(n+1,t) - C(n+1,t+1) &= (C(n,t/2) + C(n,t/2) + 2) \\
&\quad - (C(n,t/2) + C(n,t/2+1) + 2) \\
&= C(n,t/2) - C(n,t/2+1)
\end{aligned}
$$

If $t$ is even and $t + 1 \leq 2^{(n+1)-1}$ then $t/2 + 1 \leq 2^{n-1}$. Hence $C(n,t/2) - C(n,t/2+1) < 0$ by the inductive assumption, and so $C(n+1,t) - C(n+1,t+1) < 0$.

**Case 2.** *t is odd*:

$$
\begin{aligned}
C(n+1,t) - C(n+1,t+1) &= (C(n,(t+1)/2) + C(n,(t-1)/2) + 2) \\
&\quad - (C(n,(t+1)/2) + C(n,(t+1)/2) + 2) \\
&= C(n,(t-1)/2) - C(n,(t+1)/2)
\end{aligned}
$$

If $t$ is odd and $t+1 \leq 2^{(n+1)-1}$ then $(t+1)/2 \leq 2^{n-1}$. Hence $C(n,(t-1)/2) - C(n,(t+1)/2) < 0$ by the inductive assumption, and so $C(n+1,t) - C(n+1,t+1) < 0$.

Since Case 1 and 2 are exhaustive the inductive step holds. $\square$

## C $P(t=0)$ FOR PERCEPTRON WITH INFINITESIMAL $b$

If b is sampled uniformly from $[-\epsilon, \epsilon]$, then only if $|\langle w, x \rangle| < \epsilon$ can some $x$ be classified differently from a perceptron without a threshold bias term. The set of weight vectors which change the classification of non-zero $x$ becomes vanishingly small as $\epsilon$ goes to 0, but for $x = 0$, we have $P(\mathbf{1}(\langle w, 0 \rangle + b) = 0) = P(\mathbf{1}(\langle w, 0 \rangle + b) = 1) = 1/2$. Consider some function $f$, and define $g$ where $f \to g$ under the addition of an infinitesimal bias. Then with even probability the origin remains mapped to 0 (meaning $\mathcal{T}(g) = \mathcal{T}(f)$), or is mapped to 1 (meaning $\mathcal{T}(g) = \mathcal{T}(f) + 1$) as the rest of $f$ is unchanged, to $\mathcal{O}(\epsilon)$. As this is true of all $f$, $P(T = t)_{b \sim (-\epsilon, \epsilon)} = \frac{1}{2} P(T = t) + \frac{1}{2} P(T = t-1)$, leading to:

$$
P(T = t) = \begin{cases} 2^{-(n+1)} \text{ if } t = 0 \text{ or } t = 2^n \\ 2^{-n} \text{ otherwise} \end{cases}. \tag{9}
$$

For larger $\sigma_b$ $P(t=0)$ or $P(t=2^n)$ increases with increasing $\sigma_b$ as can be seen in Figure 3 of the main text.

## D ZIPF'S LAW IN A PERCEPTRON WITH $b = 0$

In (Valle-Pérez et al., 2018) it was shown empirically that the rank plot for a simple DNN exhibited a Zipf like power law scaling for larger ranks. Zipf's law occurs in many branches of science (and probably for many reasons). In this section we check whether this scaling also occurs for the Perceptron.

In Figure 5, we compare a rank plot of the probability $P(f)$ for individual functions for the simple perceptron with $b = 0$, the perceptron, and for a one layer FCN. While all architectures have $n = 7$, the perceptrons can of course express far fewer functions. Nevertheless, both the perceptrons and the more complex FCN show similar phenomenology, with a Zipf law like tail at larger ranks (i.e. a power law).

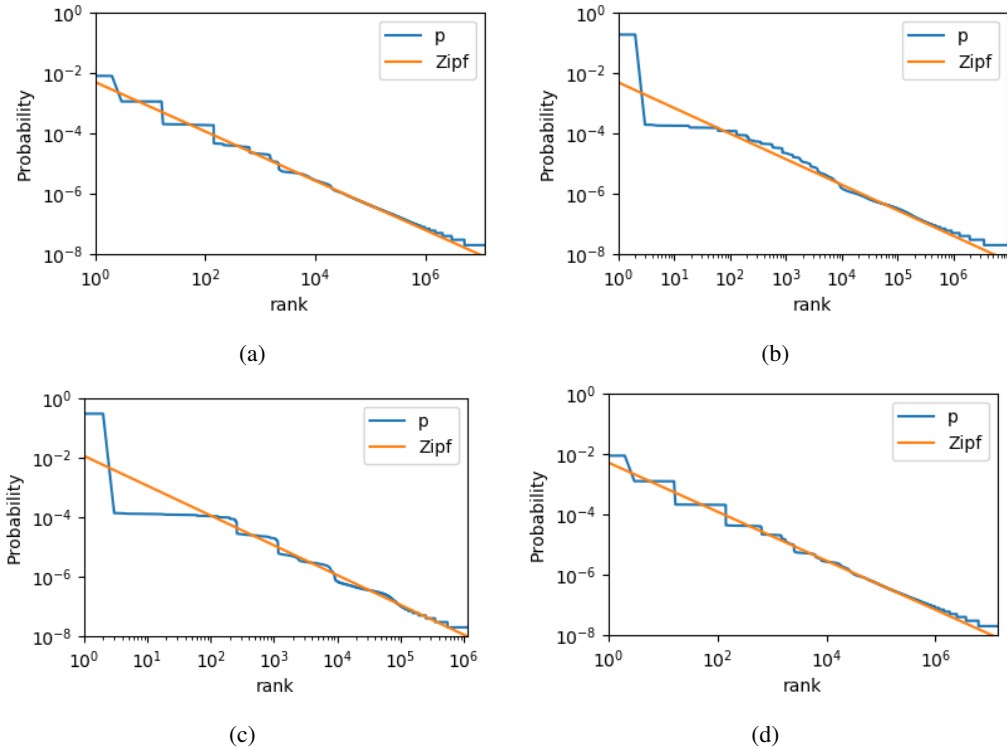

Figure 5: Probability vs rank for functions (ranked by probability) from samples of size $10^8$, with input size $n = 7$, and every weight and bias term sampled from $\mathcal{N}(0, 1)$ unless otherwise specified, over initialisations of: (a) a perceptron with $b = 0$; (b) a perceptron; (c) a one-hidden layer neural network (with 64 neurons in the hidden layer); (d) a perceptron with $b = 0$ and weights sampled from identical centered uniform distributions (note how similar (a) is to (d)!). We cut off frequencies less than 2 to eliminate finite size effects. In (a) and (b) lines were fitted using least-squares regression; for (c) the line corresponding to the ansatz in Equation (10) is plotted instead.

While the orginal formulations for Zipf's law only allows for a powerlaw with exponent 1, in practice the terminology of Zipf's law is used for other powers, such that $p = b \times \text{rank}^{-a}$ for some positive $a, b$. If we assume that this scaling persists, then we can relate the total number of functions a perceptron can express, to the constant $b$, because the total probability must integrate to 1.

For the simplest case with $a = 1$, this leads to an equation for the probability as function of rank given by

$$P(r) = \frac{1}{\ln(N_O)r},$$  (10)

where $N_0$ is the total number of functions expressible.

The FCN appears to show such simple scaling. And as the FCN of width 64 is fully expressive (see Lemma 5.1) , there are $N_O = 2^{2^7} \approx 3 \times 10^{38}$ possible Boolean functions. We plot the Zipf law prediction of Equation (10) next to the empirically estimated probabilities in Figure 5c. We observe that the curve is described well at higher values of the rank Zipf's law. Note that the mean probability uniformly sampled over functions for this FCN is $\langle P(f) \rangle = 1/N_O \approx 3 \times 10^{-39}$ so that we only measure a tiny fraction of the functions with extremely high probabilities, compared to the mean. Also, most functions have probabilities less than the mean, and only order $2^{-n}$ have probability larger than the mean. A least-squares linear fit on the log-log graph was consistent within experimental error for the ansatz.

For the perceptron with a bias term Figure 5b, we observe that the gradient differs substantially from $-1$, and a linear fit gives $\log_{10}(p) = -0.85 \log_{10}(rank) - 2.32$. Using the same arguments for calculating $N_O$ as made in Equation (10), we obtain a prediction of $N_0 = 7.63 \times 10^9$, which is 91%

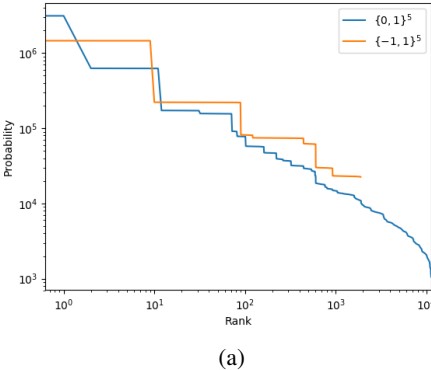 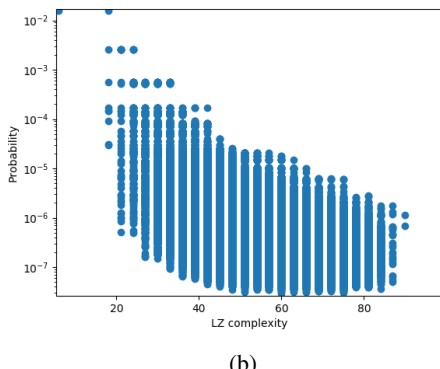

(a)                              (b)

Figure 6: (a) Probability of functions versus their rank (ranked by probability) for a perceptron with $n = 5$, and weights sampled i.i.d. from a Gaussian and no threshold bias term, acting on either centered $\{-1,1\}^5$ or uncentered $\{0,1\}^5$ data. (b) Probability of functions versus their LZ complexity for a perceptron with $n = 6$, and weights sampled i.i.d. from a Gaussian and no threshold bias term, acting on the centered Boolean hypercube $\{-1,1\}^5$.

of the known value[4]. A linear fit for the perceptron with no threshold bias term gives $\log_{10}(p) = -0.81 \log_{10}(rank) - 2.31$, leading to a prediction of $N_0 = 3.97 \times 10^8$. which is, as expected, significantly lower than a perceptron with no threshold bias term. We expect there to be some discrepancy between the pure Zipf law prediction, and the true $N_O$, because the probability seems to deviate from the Zipf-like behaviour at the highest rank, which we observe for $n = 5$ in Figure 6a, as in this case the number of functions is small enough that it becomes feasible to sample all of them.

It is also worth mentioning that a rank-probability plot for a perceptron with weights sampled from a uniform distribution (Figure 5d) is almost indistinguishable from the Gaussian case (Figure 5b), which is interesting, because when plotted against LZ complexity, as in Figure 1 of the main text, there is a small but discernible difference between the two types of initialisation.

Finally, in Figure 6a we compare the rank plot for centered and uncentered data for a smaller $n = 5$, $\sigma_b = 0$ perceptron where we can find all functions. Note that for the centered data, only functions with $t = 16$ can be expressed, which is somewhat peculiar, and of course means significantly less functions. Nevertheless, this systems still has clear bias within this one entropy class, and this bias correlates with the LZ complexity (Figure 6b) as also observed for the perceptron with centered data.

## E  FURTHER RESULTS ON THE DISTRIBUTION WITHIN $\mathbb{F}_t$

In this appendix we will denote the output function of the perceptron evaluated on $\{0,1\}^n$ by way of a bit string $f$, whose $i$'th bit is given by

$$f_i = \mathbf{1}(\langle w, bin(i)\rangle) \tag{11}$$

where $bin(i)$ takes an integer $i$ and maps it to a point $x \in \{0,1\}^n$ according to its binary representation (so $bin(5) = (1,0,1)$ and $bin(1) = (0,0,1)$ when $n = 3$).

### E.1  EMPIRICAL RESULTS

We sample $10^8$ initialisations of the perceptron, divide the list of functions into $\mathbb{F}_t$, and present probability-complexity plots for several values of $t$ in Figure 7. We use the Lempel-Ziv complexity (Lempel & Ziv, 1976; Dingle et al., 2018) of the output bit string as the approximation to the Kolmogorov complexity of the function (Dingle et al., 2018). As in (Dingle et al., 2018; Valle-Pérez et al., 2018), this complexity measure is denoted $K_{LZ}(f)$. To avoid finite size effects (as noted in (Valle-Pérez et al., 2018)), we cut off all frequencies less than or equal to 2.

---
[4]https://oeis.org/A000609/list

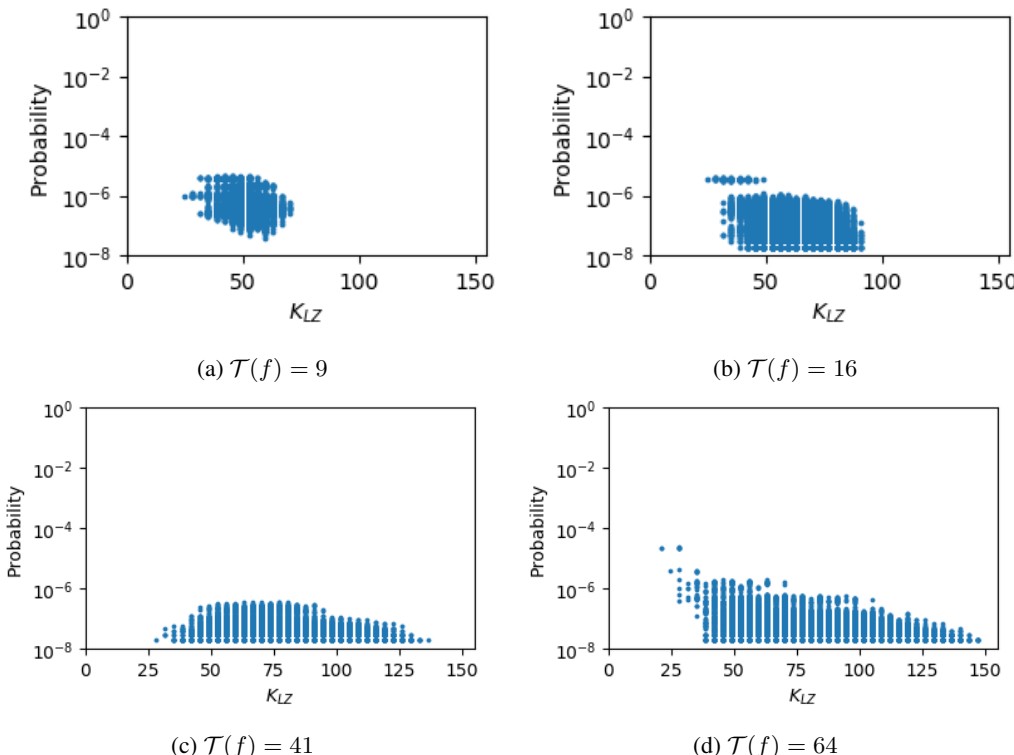

Figure 7: $P(f)$ vs $K_{LZ}(f)$ at a selection of values of $\mathcal{T}(f)$, for a perceptron with input dimension 7, weights sampled from $\mathcal{N}(0,1)$ and no threshold bias terms. We observe a large range in $K_{LZ}(f_t)$ for $f_t \in \mathbb{F}_t$ which increases as $t$ approaches 64, which is to be expected – for example the function $f = 0101\ldots$ is very simple and has maximum entropy, and we expect there to exist higher complexity functions at higher entropy. Consistent with the bound in (Valle-Pérez et al., 2018), simpler functions tend to have higher probabilities than more complex ones. The data-points at especially high probabilities in Figure 7d correspond to the function $f = 0101\ldots$ and equivalent strings after permuting dimensions.

As can be seen in Figure 1 the probability-complexity graph satisfies the simplicity bias bound Equation (1) for all functions. Now, in Figure 7) we observe subsets $\mathbb{F}_f$ of the overall set of functions. Firstly, we observe, as expected, that smaller $t$ means a smaller range in $K_{LZ}$, since high complexity functions are not possible at low entropy. Conversely, low complexity functions are possible at high entropy (say for 010101...), and so a larger range of probabilities and complexities is observed for $t = 64$. For these larger entropies, an overall simplicity bias within the set of fixed $t$ can be observed.

The larger range observed in $t = 64$ and $t = 16$ (compared to $t = 41$ and $t = 9$) can be explained by the presence of highly ordered functions having those $t$ values - for example, in $t = 64$, there is $f = 0101\ldots$ and its symmetries; and in $t = 16$ there are functions such as $f = 00010001\ldots$ and its symmetries. The other two $t$ values do not divide $2^7$, so there will be no functions with such low block entropy (implying low $K_{LZ}$).

We also demonstrate differences in the variation in $P(f)$ vs $K_{LZ}(f)$ when $w$ is sampled from uniform distributions, in Figure 8, and compare these pots to those in Figure 7. Whilst we know from Theorem 4.1 that sampling $w$ from a uniform distribution will not affect $P(t)$, it is not hard to see that there will be some variation in function probability within the classes $\mathbb{F}_t$. We observe that the simple functions which have high probability for $t = 64$ when the perceptron is initialised from a Gaussian (Figure 7d) have lower probabilities in the uniform case (Figure 8d). We comment further on this behaviour in Appendix E.2. However, we see limited differences in their respective rank-probability plots ( Figure 5a and Figure 5d).

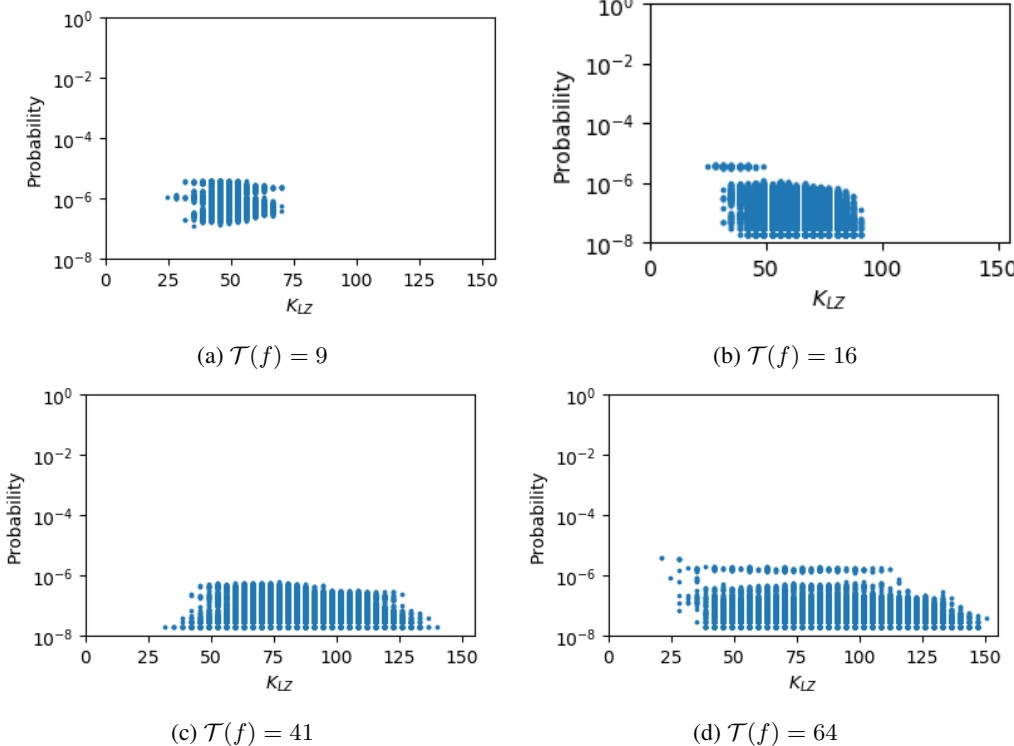

Figure 8: $P(f)$ vs $K_{LZ}(f)$ at a selection of values of $\mathcal{T}(f)$, for a perceptron with input dimension 7, weights sampled from a centered uniform distribution and no threshold bias terms. We compare to Figure 7, and observe that uniform sampling reduces slightly reduces the simplicity bias within the sets $\mathbb{F}_t$ (see Appendix E.2).

### E.2 SUBSTRUCTURE WITHIN $\{0, 1\}^n$

Consider a subset $\mathcal{H}^m \subset \{0, 1\}^n$ such that $0 \in \mathcal{H}^m$. We have $\binom{n}{m}$ such subsets. Then the marginal distribution over $\mathcal{H}^m$ is given by

$$P\left(\sum_{x \in \mathcal{H}^m} \mathbf{1}(\langle w, x \rangle) = t\right) = 2^{-m} \tag{12}$$

This is again independent of the distribution of $w$ (provided it's symmetric about coordinate planes). We give two example applications of Equation (12) in Equation (13). We use $*$ to mean any allowed value, and sum over all allowed values,

$$\sum_* P(f = "0 * 0 * \ldots 0 * 0 *") = 2^{-(n-1)}$$

$$\sum_* P(f = "0 * * * \ldots 0 * * *") = 2^{-(n-2)} \tag{13}$$

We can apply the same argument that we applied in Section 4 to any set of bits in $f$ defined by some $\mathcal{H}^m$, to show that there is an "entropy bias" within each of these substrings. However, these identities imply a strong bias within each $\mathbb{F}_t$. For the case of full expressivity, assuming each value of $t$ has probability $2^{-n}$, and every string with the same value of $t$ is equally likely, one gets probabilities very close to those in Equation (13) (although slightly lower). However, the perceptron is not fully expressive, so it is unclear how much the probabilities on Equation (13) are due purely to the entropy bias, and how much is due to bias within each $\mathbb{F}_t$.

It is difficult to calculate the exact probabilities of any function for Gaussian initialisation[5]. It may not be possible to fine-grain probabilities analytically further than Equation (12) (although we can use the techniques in Section 4.3 to come up with analytic expressions for $P(f)$ for all $f$).

However, we can calculate some probabilities quite easily when $w$ is sampled from a uniform distribution, $w \sim \mathcal{U}^n(-1, 1)$. By way of example, we calculate $P(f = \widetilde{f})$ for $\widetilde{f} = "0101 \ldots 0101"$. The conditions for $\widetilde{f}$ are as follows: $w_n > 0$, $w_i < 0 \; \forall i \neq n$ and $\sum_j w_j > 0$, so

$$P(f = \widetilde{f}) = 2^{-n} \int_0^1 \frac{x^{n-1}}{(n-1)!} dx = \frac{2^{-n}}{n!} \tag{14}$$

Using Equation (3), we can calculate how much more likely the function is than expected,

$$\frac{P(f = \widetilde{f})}{\langle P(f_{t=2^n-1}) \rangle} = \frac{|F_{t=2^n-1}|}{n!} \tag{15}$$

For $n = 5$, we can calculate Equation (15) using[6] $|F_{t=2^n-1}| = 370$ and $w \sim \mathcal{U}^n(-1, 1)$ and obtain $P(f = \widetilde{f})/\langle P(f_{t=2^n-1}) \rangle = 3.08$. This clearly shows that $\widetilde{f}$ is significantly more likely than expected just by using results from Equation (3). Empirical results further suggest that for Gaussian initialisation of $w$, $P(f = \widetilde{f})/\langle P(f_{t=2^n-1}) \rangle \approx 10$. This, plus data in Appendix E.1 suggest that Gaussian initialisation may lead to more simplicity bias.

## F  TOWARDS UNDERSTANDING THE SIMPLICITY BIAS OBSERVED WITHIN $\mathbb{F}_t$

Here we offer some intuitive arguments that aim to explain why there should be further bias towards simpler functions within $\mathbb{F}_t$.

We first need several definitions.

We define a set $\mathbb{A} \subset R_{\geq 0}^n$ such that $a \in \mathbb{A}$ iff $a_i < a_j \; \forall i < j$, interpreted as the absolute values of the weight vector.

We now define four sets, $\Gamma \; \Sigma$ and $\Upsilon$, which classify the types of linear conditions on the weights of a perceptron acting on an $n$-dimensional hypercube:

1. $\Gamma$, the set of permutations in $\{1, \ldots, n\}$ (which we can interpret as permutations of the axes in $\mathbb{R}^n$ or as possible orders of the absolute values of the weights if no two of them are equal).
2. $\Sigma = \{-1, +1\}^n$ (which is interpreted as the signature of the weight vector)
3. $\Upsilon$ is the set of linear inequality conditions on components of $a \in \mathbb{A}$, which include more than two elements of $a$ (so they exclude the conditions defining $\mathbb{A}$.

A unique weight vector can be specified giving its signature $\sigma \in \Sigma$, the order of its absolute values $\gamma \in \Gamma$, and a value of $a \in \mathbb{A}$. On the other hand, a unique *function* can be specified by a set of linear conditions on the weight vector $w$. These conditions can be divided into three types: $\Sigma$ (signs), $\Gamma$ (ordering), and $\Upsilon$ (any other condition).

The intuition to understand the variation in complexity and probability over different functions is the following. Each function corresponds to a unique set of necessary and sufficient conditions on the weight vector (corresponding to the faces of the cone in weight space producing that function). We argue that different functions have conditions which vary a lot in complexity, and we conjecture that this correlates with their probability, as we discuss in Section 4.3 in the main text.

As a first approach in understanding this, we consider the role of symmetries under permutations of dimensions. Any string that is symmetric under a permutation of $k$ dimensions[7], can't have necessary conditions that represent relative orderings of those $k$ dimensions. Furthermore the set of

---

[5]Except for $f_{t<3}$, because we know all functions within constant $t$ for $t < 3$ are equivalent under permutation of the dimensions so all their probabilities are equal to the average probabilities given by Equation (3)

[6]370 unique functions were obtained by sampling $10^{10}$ weight vectors so this is technically a lower bound

[7]or example, "0101010..." is symmetric under permutation of $n - 1$ dimensions

conditions must be invariant under these permutations. This strongly constraints the sets of conditions that highly symmetric strings like "010101..." or "11111...00000..." can have, to be relatively simple sets.

We now consider the set of necessary conditions in $\Upsilon$ for different functions. We expect that conditions in $\Upsilon$ are more complex than those in $\Gamma$ and $\Sigma$. Furthermore, we find that functions have a similar number of minimal conditions[8], so that more conditions in $\Upsilon$ seems to imply fewer conditions in $\Gamma$ and $\Sigma$, and therefore, a more complex set of conditions overall.

We are going to explicitly study the functions within each set of constant $t$, for some small values of $t$, and find their corresponding conditions in $\Upsilon$ and $\Sigma$. We fix $\Gamma$ to be the identity for simplicity. The conditions for a particular function should include the conditions we find here plus their corresponding conditions under any permutation of the axes which leaves the function unchanged. This means that on top of the describing the conditions for a fixed $\Gamma$, we would need to describe the set of axes which can be permuted. This will result in a small change to the Kolmogorov complexity of the set of conditions, specially small for functions with many symmetries or very few symmetries.

We find that the conditions appear to be arranged in a decision tree (a consequence of Theorem 4.1) with a particular form. First, we will prove the following simple lemma which bounds the value of $t$ that is possible for some special cases of $\sigma$.

**Lemma F.1.** *We define $t_{min}(\sigma)$ to be the minimum possible value of $\mathcal{T}(\sigma \odot a)$ for fixed $\sigma$ over all $a$. We define $t_{max}$ similarly. Consider $\sigma = (\underbrace{-1, \ldots, -1}_{k-1}, +1, \underbrace{-1, \ldots, -1}_{n-k})$. Then:*

1. *$t_{max}(\sigma) = 2^{k-1}$*
2. *$t_{min}(\sigma) = k$*
3. *Changing $\sigma_i = \{+1\}$ for some $i < k$ will lead to an increase in $t_{min}(\sigma)$*

*Proof.* 1. For any $a$, $f(x) = 1$ for all (exactly $k$) points $x \in \{0, 1\}^n$ which satisfy

$$(x_k = 1) \text{ and } (x_i = 1 \text{ for exactly one } i \leq k) \text{ and } (x_j = 0 \text{ if } j \neq i, k)$$

We can set $f(x) = 0$ for all other $x \in \{0, 1\}^n$ by imposing the condition $a_1 + a_2 > a_k$. Thus $t_{min}(\sigma) = k$.

2. We can restrict the values of $a_i$ for $i < k$ to be arbitrarily smaller than $a_k$ provided they satisfy the ordering condition on $a$, and thus if we impose the condition

$$\sum_{i=1}^{i=k-1} a_i < a_k$$

then for any $a$, $f(x) = 1$ for all $x \in \{0, 1\}^n$ which satisfy

$$(x_k = 1) \text{ and } (x_l = 0 \text{ if } l > k)$$

We can see that, for all $a$, $f(x) = 0$ for any $x$ which does not satisfy these conditions, because $a_k < a_l$ for all $k < l$ and $\sigma_k$ is the only positive element in $\sigma$. Thus $t_{max} = 2^{k-1}$

3. On changing $\sigma$ such that $\sigma_i = \{+1\}$, all $x \in \{0, 1\}^n$ which previously satisfied $f(x) = 1$ will remain mapped to 1, plus at least the one-hot vector with $a_i = 1$. $\qquad \square$

We will now sketch a procedure that allows one to enumerate the conditions in $\Upsilon$ and $\Sigma$ (corresponding to conditions on $a$ and on $\sigma$ respectively) such that they satisfy $\mathcal{T}(\sigma \odot a) = t$, for any given $t$.

1. From Lemma F.1 we see that all $\sigma_i = -1$ for $i > t$ in order for $t_{min}(\sigma) \leq t$.

2. Iterate through all $2^t$ distinct $\sigma$ which satisfy 1., and retain only those which also satisfy $t_{min}(\sigma) \leq t$. $t_{min}(\sigma)$ can be computed by counting the number of inputs $x \in \{0, 1\}^n$ such that for every $x_i = 1$ with $\sigma_i = -1$, there exists a $x_j$ with $j > i$ such that $\sigma_j = 1$. These are all the $x$ such that $\sigma$ and $a$ imply they are mapped to 1 without further conditions on $a$.

---

[8]In fact we conjecture that the number of conditions is close to $n$ for most functions, although we empirically find that it can sometimes be larger too

3. Find conditions on these $\sigma$ such that $\mathcal{T}(\sigma \odot a) = t$. We can find the possible values of $\mathcal{T}(\sigma \odot a)$ by first ordering the $x$ in a way that satisfies $x < x'$ if $x$ has less 1s than $x'$. We then traverse the decision tree corresponding to mapping each of the $x$, in order, to either 1 or 0. At each step, we propagate the decision by finding all $x$ not yet assigned an output, which can be constructed as a linear combination of $x$'s with assignments, where the coefficients in the combination have opposite signs for $x'$ mapped to different outputs. We stop the traversal if at some point more than $t$ points are mapped to 1. Each of the decisions in the tree, correspond to a new condition on $a$. We denote these conditions by $\upsilon$.

For small values of $t$ one can perform this search by hand. For example, we consider $t = 4$. We find that all signatures with $\sigma_{i>4} = -1$ are the only set that have $t_{min} \leq 4$. As an example we consider the signature $\sigma = \{+1, +1, -1, ..., -1\}$. For this signature to result in $\mathcal{T}(\sigma \odot a) = 4$, we need conditions on $x_1 = (1, 1, 0, 1 \dots)$ and $x_2 = (1, 1, 1, 0 \dots)$ which are $(a_4 > a_1 + a_2)$ and $(a_3 < a_1 + a_2)$. Figure 9 shows the full sets for $t = 4$ and $t = 5$. We observe that each branching corresponds to complementary conditions - which is to be expected, as there exists a signature producing $t$ for any $a$, as per Theorem 4.1.

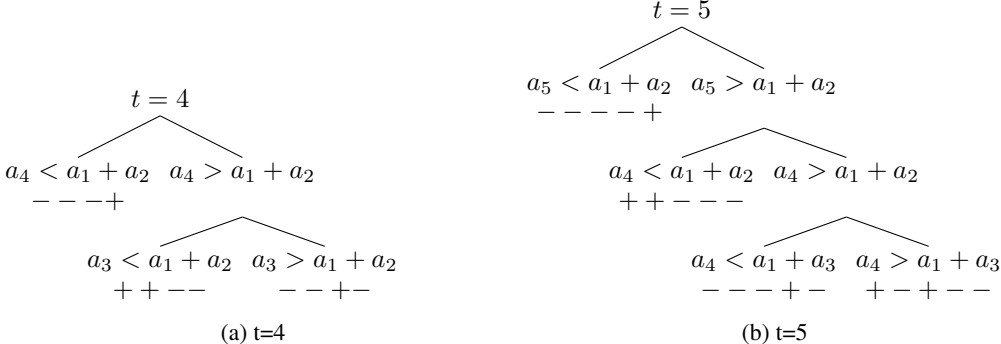

(a) t=4          (b) t=5

Figure 9: We assume that the signs to the right of those shown are all negative. We can list the various non-equivalent classes of functions and their conditions in a pictorial form. A condition at any node in the graph is also a condition for any daughter node - by way of example, we would read the conditions of $t = 4$ for the sign arrangement $++--$ (see Figure 9a) as $((a_4 > a_1 + a_2) \cap (a_3 < a_1 + a_2))$.

Each distinct condition on $a$ and $\sigma$ produces a unique function $f$, as the constructive procedure above produces the inequalities on $a_i$ by specifying the outputs of each input not already implied by the set of conditions, thus uniquely specifying the output of every input. We now show that there is a large range in the number of conditions in $\Upsilon$ required to specify different function. First, as the analysis in the proof of Lemma F.1 shows, for every $t$ there exists a signature which only requires one further condition,

$$\sigma = (\underbrace{-1, \dots, -1}_{t-1}, +1, \underbrace{-1, \dots, -1}_{n-t}), \ \upsilon = a_t < a_1 + a_2$$

Furthermore, we prove in Lemma F.2 that for all $n$ there is at least one function which has $n - 2$ conditions in $\Upsilon$ (which are neither in $\Sigma$ or $\Gamma$). This implies that there is a large range in the number of conditions in $\Upsilon$ for large $n$.

**Lemma F.2.** *There exists a function with $n-2$ inequalities in $\Upsilon$, that is not including those induced by $\Sigma$ or in $\Gamma$.*

*Proof.* We prove this by induction.

Base case $n = 2$: There exists a function with minimal conditions corresponding to $(1, 0)$ mapping to 1 and $(1, 1)$ mapping to 0.

Assume that in $n$ dimensions, there exists a function with $n$ minimal conditions corresponding to points $\{p_i\}_{i=1}^n$ with $p_i = (1, \dots, 1, 0, \dots, 0)$ with 1s in the first $i$ positions, and 0 in the last $n - i$

positions, being mapped to $f(p_i) = (1 + (-1)^{i+1})/2$. Now, in $n + 1$ dimensions, we can extend each of those $n$ conditions (planes) by adding a 0 to the $n$th coordinate of each point $p_i$. We now consider a cone in $n$ dimensions bounded by these planes and bounded by the $w_{n+1} = 0$ plane, with $w_{n+1} > 0$ if $n$ is even or $w_{n+1} < 0$ if $n$ is odd. If $n$ is even, we can keep increasing $w_{n+1}$ until we cross the plane $p_{n+1} = 0$, with $p_{n+1} = (1, \ldots, 1)$. We do the same, decreasing $w_{n+1}$ if $n$ is odd. After we crossed the $p_{n+1} = 0$ plane, the boundaries of the plane will be $\{p_i\}_{i=1}^{n+1}$ with $p_i = (1, \ldots, 1, 0, dots, 0)$ with 1s in the first $i$ positions, and 0 in the last $n + 1 - i$ positions, finishing the induction.

$\square$

We conjecture that more conditions in $\Upsilon$ (i.e. more complex conditions) correlates with lower probability (if $w$ is sampled from a Gaussian distribution). If this is true for higher $t$, we should expect to see high probabilities correlating with lower complexities, which if true would explain the "simplicity bias" we observe beyond the entropy bias.

## G  EXPRESSIVITY CONDITIONS

**Lemma 5.1.** *A neural network with layer sizes $\langle n, 2^{n-1}, 1 \rangle$ can express all Boolean functions over $n$ variables.*

*Proof.* Let $f : \{0, 1\}^n \to \{0, 1\}$ be any Boolean function over $n$ variables, and let $t$ be the number of vectors in $\{0, 1\}^n$ that $f$ maps to 1. Let $\overline{f}$ be the negation of $f$, that is, $\overline{f}(x) = 1 - f(x)$.

It is possible to express $f$ as a Boolean formula $\phi$ in Disjunctive Normal Form (DNF) using $t$ clauses, and $\overline{f}$ as a Boolean formula $\psi$ using $2^n - t$ clauses (see Appendix B). Let $\phi$ and $\psi$ be specified over the variables $x_1 \ldots x_n$.

We can specify a neural network $N_\phi = \langle W_{\phi 1}, b_{\phi 1}, W_{\phi 2}, b_{\phi 2} \rangle$ with layer sizes $\langle n, t, 1 \rangle$ that expresses $f$ by mimicking the structure of $\phi$. Let $W_{\phi 1[i,j]} = 1$ if $x_i$ is positive in the $j$'th clause of $\phi$, $W_{\phi 1[i,j]} = -1$ if $x_i$ is negative in the $j$'th clause of $\phi$, and $W_{\phi 1[i,j]} = 0$ if $x_i$ does not occur in the $j$'th clause of $\phi$ (note that if the construction in Appendix B is followed then every variable occurs in every clause in $\phi$). Let $b_{\phi 1[j]} = 1 - \gamma$, where $\gamma$ is the number of positive literals in the $j$'th clause of $\phi$, let $b_{\phi 2} = 0$, and let $W_{\phi 2[j]} = 1$ for all $j$. Now $N_\phi$ computes $f$. Intuitively every neuron in the hidden layer corresponds to a clause in $\phi$, and the output neuron corresponds to an OR-gate.

We can similarly specify a neural network $N_\psi = \langle W_{\psi 1}, b_{\psi 1}, W_{\psi 2}, b_{\psi 2} \rangle$ with layer sizes $\langle n, 2^n - t, 1 \rangle$ that expresses $\overline{f}$ by mimicking the structure of $\psi$. Using this network we can specify a third network $N_\theta = \langle W_{\theta 1}, b_{\theta 1}, W_{\theta 2}, b_{\theta 2} \rangle$ with the same layer sizes as $N_\psi$ that expresses $f$ by letting $W_{\theta 1} = W_{\psi 1}, b_{\theta 1} = b_{\psi 1}, W_{\theta 2} = -W_{\psi 2}$, and $b_{\theta 2} = -b_{\psi 2} + 0.5$. Intuitively $N_\theta$ simply negates the output of $N_\psi$.

Therefore, any Boolean function $f$ over $n$ variables is expressed by a neural network $N_\phi$ with layer sizes $\langle n, t, 1 \rangle$ and by a neural network $N_\theta$ with layer sizes $\langle n, 2^n - t, 1 \rangle$. Since either $t \leq 2^{n-1}$ or $2^n - t \leq 2^{n-1}$ it follows that any Boolean function over $n$ variables can be expressed by a neural network with layer sizes $\langle n, 2^{n-1}, 1 \rangle$. $\square$

**Lemma 5.2.** *A neural network with $l$ hidden layers and layer sizes $\langle n, (n + 2^{n-1-\log_2 l} + 1), \ldots (n + 2^{n-1-\log_2 l} + 1), 1 \rangle$ can express all Boolean functions over $n$ variables.*

*Proof.* Let $f : \{0, 1\}^n \to \{0, 1\}$ be any Boolean function over $n$ variables, and let $t$ be the number of vectors in $\{0, 1\}^n$ that $f$ maps to 1. Let $\overline{f}$ be the negation of $f$, that is, $\overline{f}(x) = 1 - f(x)$.

It is possible to express $f$ as a Boolean formula $\phi$ in Disjunctive Normal Form (DNF) using $t$ clauses, and $\overline{f}$ as a Boolean formula $\psi$ using $2^n - t$ clauses (see Appendix B). Let $\phi$ and $\psi$ be specified over the variables $x_1 \ldots x_n$.

Let $N_\phi = \langle W_{\phi 1}, b_{\phi 1}, \ldots W_{\phi l+2}, b_{\phi l+2} \rangle$ be a neural network with layer sizes $\langle n, (n + \lceil t/l \rceil + 1), \ldots, (n + \lceil t/l \rceil + 1), 1 \rangle$.

For $2 \leq k \leq l$:

- Let $W_{\phi k A}$ be an identity matrix of size $n \times n$, and $b_{\phi k A}$ a zero vector of length $i$.

- Let $W_{\phi k B}$ be a matrix of size $n \times \lceil t/l \rceil$ such that $W_{\phi k B[i,j]} = 1$ if $x_i$ is positive in the $((k-1) \times (\lceil t/l \rceil) + j)$'th clause of $\phi$, $-1$ if it is negative, and $0$ if $x_i$ does not occur in the $((k-1) \times (\lceil t/l \rceil) + j)$'th clause of $\phi$ (note that if the construction in Appendix B is followed then every variable occurs in every clause in $\phi$). Let $b_{\phi k B}$ be a vector of length $\lceil t/l \rceil$ such that $b_{\phi k B[j]} = 1 - \gamma$, where $\gamma$ is the number of positive literals in the $((k-1) \times (\lceil t/l \rceil) + j)$'th clause of $\phi$.

- Let $W_{\phi k C}$ be a unit matrix of size $(\lceil t/l \rceil + 1) \times 1$, and let $b_{\phi k C} = [0]$.

- Let $W_{\phi k}$ and $b_{\phi k}$ be the concatenation of $W_{\phi k A}$, $W_{\phi k B}$, $W_{\phi k C}$ and $b_{\phi k A}$, $b_{\phi k B}$, $b_{\phi k C}$ respectively in the following way:

$$W_{\phi k} = \begin{bmatrix} [W_{\phi k A}] & [W_{\phi k B}] & 0_{(\lceil t/l \rceil + 1) \times 1} \\ 0_{n \times n} & 0_{n \times \lceil t/l \rceil} & [W_{\phi k C}] \end{bmatrix}, b_{\phi k} = [[b_{\phi k A}] \quad [b_{\phi k B}] \quad [b_{\phi k C}]]$$

Let $W_{\phi 1}$ and $b_{\phi 1}$ be the concatenation of $W_{\phi k A}$, $W_{\phi k B}$, and $b_{\phi k A}$, $b_{\phi k B}$ respectively, where these are constructed as above. Let $W_{\phi(l+2)}$ be a unit matrix of size $(\lceil t/l \rceil + 1) \times 1$, and let $b_{\phi(l+2)} = [0]$.

Now $N_\phi$ computes $f$. Intuitively each hidden layer is divided into three parts; $n$ neurons that store the value of the input to the network, $\lceil t/l \rceil$ neurons that compute the value of some of the clauses in $\phi$, and one neuron that keeps track of whether some clause that has been computed so far is satisfied.

We can similarly specify a neural network $N_\psi = \langle W_{\psi 1}, b_{\psi 1}, \ldots W_{\psi l+2}, b_{\psi l+2} \rangle$ with layer sizes $\langle n, (n + \lceil (2^n - t)/l \rceil + 1), \ldots, (n + \lceil (2^n - t)/l \rceil + 1), 1 \rangle$ that expresses $\overline{f}$. Using this network we can specify a third network $N_\theta = \langle W_{\theta 1}, b_{\theta 1}, \ldots W_{\theta l+2}, b_{\theta l+2} \rangle$ with the same layer sizes as $N_\psi$ that expresses $f$ by letting $W_{\theta k} = W_{\psi k}, b_{\theta k} = b_{\psi k}$ for $k \leq l$ and $W_{\theta(l+2)} = -W_{\psi(l+2)}$, and $b_{\theta(l+2)} = -b_{\psi(l+2)} + 0.5$. Intuitively $N_\theta$ simply negates the output of $N_\psi$.

Therefore, any Boolean function $f$ over $n$ variables is expressed by a neural network $N_\phi$ with layer sizes $\langle n, (n + \lceil t/l \rceil + 1), \ldots, (n + \lceil t/l \rceil + 1), 1 \rangle$ and by a neural network $N_\theta$ with layer sizes $\langle n, (n + \lceil (2^n - t)/l \rceil + 1), \ldots, (n + \lceil (2^n - t)/l \rceil + 1), 1 \rangle$. Since either $t \leq 2^{n-1}$ or $2^n - t \leq 2^{n-1}$ it follows that any Boolean function over $n$ variables can be expressed by a neural network with layer sizes $\langle n, (n + \lceil 2^{n-1}/l \rceil + 1), \ldots, (n + \lceil 2^{n-1}/l \rceil + 1), 1 \rangle = \langle n, (2^{n-1-\log_2 l} + n + 1), \ldots (2^{n-1-\log_2 l} + n + 1), 1 \rangle$. $\square$

## H  THEOREMS ASSOCIATED WITH ENTROPY INCREASE FOR DNNS

We define the data matrix for a general set of input points below.

**Definition H.1** (Data matrix). *For a general set of inputs, $\{x^{(i)}\}_{i=1,\ldots,m}$, $x_i \in \mathbb{R}^n$, we define the data matrix $\boldsymbol{X} \in \mathbb{R}^{m \times n}$ which has elements $X_{ij} = x^{(i)}_j$, the $j$-th component of the $i$-th point.*

**Lemma 5.3.** *For any set of inputs $\mathbb{S}$, the probability distribution on $T$ of a fully connected feedforwad neural network with linear activations, no bias, and i.i.d. initialisation of the weights is equivalent to an perceptron with no bias and i.i.d. weights.*

*Proof.* Consider a neural network with $L$ layers and weight matrices $w_0 \ldots w_L$ (notation in Section 5) acting on the set of points $\mathbb{S}$. The output of the network on an input point $x \in \mathbb{S}$ equals $\widetilde{w}x$ where $\widetilde{w} = w_L w_{L-1} \ldots w_1 w_0$. As the weight matrices $w_i$ are i.i.d., their distributions are spherically symmetric $P(w_i = aR) = P(w_i = a)$ for any rotation matrix $R$ in $\mathbb{R}^{n_i}$ and matrix $a \in \mathbb{R}^{n_{i+1} \times n_i}$. This implies that $P(\widetilde{w} = aR) = P(\widetilde{w} = a)$. Because the value of $T$ is independent of the magnitude of the weight, $|\widetilde{w}|$, this means that $P(T = t)$ is equivalent to that of an perceptron with i.i.d. (and thus spherically symmetric) weights. $\square$

One can make Lemma 5.3 stronger, by only requiring the first layer weights $w_0$ to have a spherically symmetric distribution, and be independent of the rest of the weights. If the set of inputs is the

hypercube, $\mathbb{S} = \mathcal{H}^n$, one needs even weaker conditions, namely that the distribution of $w_0$ is symmetric under reflections along the coordinate axes (as in Theorem 4.1) and has signs independent of the rest of the layer's weights. This implies that the condition of Theorem 4.1 is satisfied by $\widetilde{w}$.

**Lemma 5.4.** *Applying a ReLU function in between each layer produces a lower bound on $P(T = 0)$ such that $P(T = 0) \geq 2^{-n}$.*

*Proof.* Consider the action of a neural network $\langle n, l_1, \ldots, l_p, 1 \rangle$ with ReLU activation functions on $\{0, 1\}^n$. After passing $\{0, 1\}^n$ through $l_1 \ldots l_p$ and applying the final ReLU function after $l_p$, all points must lie in $\mathbb{R}^n_{\geq 0}$ by the definition of the ReLU function. Then, if $w$ is sampled from a distribution symmetric under reflection in coordinate planes:

$$2^{-n} = P(w \cdot \mathbb{R}^n_{\geq 0} \leq 0) \leq P(T = 0, ReLU)$$

This result also follows from Corollary H.7.

We observe $P(T = 2^n - 1) \approx P(T = 0)$ because the two states are symmetric except in cases where multiple points are mapped to the origin. This happens with zero probability for infinite width hidden layers. $\qquad\square$

**Theorem 5.5** *Let $\mathbb{S}$ be a set of $m = |\mathbb{S}|$ input points in $\mathbb{R}^n$. Consider neural networks with i.i.d. Gaussian weights with variances $\sigma_w^2/\sqrt{n}$ and biases with variance $\sigma_b$, in the limit where the width of all hidden layers $n$ goes to infinity. Let $N1$ and $N2$ be such a neural networks with $L$ and $L + 1$ infinitely wide hidden layers, respectively, and no bias. Then, the following holds: $\langle H(T) \rangle$ is smaller than or equal for $N2$ than for $N1$. It is strictly smaller if there exist pairs of points in $\mathbb{S}$ with correlations less than 1. If the networks has sufficiently large bias ($\sigma_b > 1$ is a sufficient condition), the result still holds. For smaller bias, the result holds only for sufficiently large number of layers $L$.*

*Proof.* The covariance matrix of the activations of the last hidden layer of a fully connected neural network in the limit of infinite width has been calculated and are given by the following recurrence relation for the covariance of the outputs[9] at layer $l$ [[cite]]:

$$K^l(x, x') = \sigma_b^2 + \frac{\sigma_w^2}{2\pi}\sqrt{K^{l-1}(x, x)K^{l-1}(x', x')}\left(\sin\theta_{x,x'}^{l-1} + \left(\pi - \theta_{x,x'}^{l-1}\right)\cos\theta_{x,x'}^{l-1}\right) \quad (16)$$

$$\theta_{x,x'}^l = \cos^{-1}\left(\frac{K^l(x, x')}{\sqrt{K^l(x, x)K^l(x', x')}}\right) \quad (17)$$

For the variance the equation simplifies to

$$K^l(x, x) = \sigma_b^2 + \frac{\sigma_w^2}{2}K^{l-1}(x, x) \quad (18)$$

The correlation at layer $l$ is $\rho^l(x, x') = \frac{K^l(x,x')}{\sqrt{K^l(x,x)K^l(x',x')}}$, can be obtained using Equation (16) above recursively as

$$\rho^l(x, x') = \frac{\sigma_b^2 + \frac{\sigma_w^2}{2}\sqrt{K^{l-1}(x, x)K^{l-1}(x', x')}\rho_0^l(x, x')}{\sqrt{\sigma_b^2 + \frac{\sigma_w^2}{2}K^{l-1}(x, x)}\sqrt{\sigma_b^2 + \frac{\sigma_w^2}{2}K^{l-1}(x', x')}},$$

---

[9]Note that we can speak interchangeably about the correlations at hidden layer $l - 1$, or the correlations of the output at layer $l$, as the two are the same. This is a standard result, which we state, for example, in the proof of Corollary H.7

where

$$\rho_0^l(x, x') = \frac{1}{\pi} \left( \sin \cos^{-1} \left( \rho^{l-1}(x, x') \right) + \left( \pi - \cos^{-1} \left( \rho^{l-1}(x, x') \right) \right) \rho^{l-1}(x, x') \right).$$

For the case when $\sigma_b = 0$, this simply becomes to $\rho^l(x, x') = \rho_0^l(x, x')$.

This function is 1 when $\rho^{l-1}(x, x') = 1$, and has a positive derivative less than 1 for $0 \leq \rho^{l-1}(x, x') < 1$, which implies that it is greater than $\rho^{l-1}(x, x')$. Therefore the correlation between any pair of points increases if $\rho^{l-1}(x, x') < 1$ or stays the same if $\rho^{l-1}(x, x') = 1$, as you add one hidden layer. By Lemma H.2, this then implies the theorem, for the case of no bias.

When $\sigma_b > 0$, we can write, after some algebraic manipulation

$$\frac{\rho^l(x, x')}{\rho^{l-1}(x, x')} = \frac{1 + \gamma \frac{\rho_0^l(x, x')}{\rho_0^{l-1}(x, x')}}{\sqrt{1 + \gamma \left( \frac{1}{K^{l-1}(x, x)} + \frac{1}{K^{l-1}(x', x')} \right) + \gamma^2}} \tag{19}$$

$$\geq \frac{1 + \gamma a}{\sqrt{1 + \gamma \frac{2}{\sigma_b^2} + \gamma^2}} \tag{20}$$

$$:= \phi(\gamma), \tag{21}$$

where $\gamma = \frac{\sigma_w^2 K^{l-1}(x, x')}{2 \sigma_b^2} > 0$ and $a = \frac{\rho_0^l(x, x')}{\rho_0^{l-1}(x, x')} > 1$, and the inequality follows from $K^l(x, x) \geq \sigma_b^2$ for any $x$, which follows from Equation (16).

We will study the behaviour of $\phi(\gamma)$ as a function of $\gamma$ for different values of $a$ and $\sigma_b^2$. For $\gamma = 0$, this function equals 1. If $\frac{1}{a} < \sigma_b^2 < a$, its derivative is positive, and therefore is greater than 1 for $\gamma > 0$. If $a < \sigma_b^2$, there is a unique maximum for $\gamma > 0$ at $\gamma = \frac{a \sigma_b^2 - 1}{\sigma_b^2 - a}$. Because the function tends to $a$ as $\gamma \to \infty$, if it went below 1, then at some $\gamma > 0$ it should cross 1 (by the intermediate value theorem), and by the mean value theorem, therefore it would have an extremum below one, and thus a local minimum, giving a contradiction. Thus the function is always greater than 1 when $\sigma_b > \frac{1}{a}$.

When $\sigma_b < \frac{1}{a}$, $\phi(\gamma)$ can be less than 1, thus the decreasing the correlations, for some values of $\gamma$. We know that if $\gamma \geq \frac{2(1 - a\sigma_b^2)}{(a^2 - 1)\sigma_b^2}$ the function is greater than or equal to 1. However, because of the inequality in Equation (19), we can't say what happens when $\gamma$ is smaller than this.

From Equation (18), we know that if $\sigma_w^2 \geq 2$, $K^{l-1}(x, x)$ and $K^{l-1}(x', x')$ grow unboundedly as $l$ grows. By the above arguments applied to the expression in Equation (19) before taking the inequality, this implies that after some sufficiently large $l$, $\frac{\rho^l(x, x')}{\rho^{l-1}(x, x')}$ will be $> 1$. If $\sigma_w^2 < 2$, $K^{l-1}(x, x)$ and $K^{l-1}(x', x')$ tend to $\frac{\sigma_b^2}{1 - \sigma_w^2/2}$ which also becomes the fixed point of the equation for $K^l(x, x')$, Equation (16), implying that $\rho^l(x, x') \to 1$ as $l \to \infty$, so that the correlation must increase with layers after a sufficient number of layers, and thus the moments by Lemma H.2.

Finally, applying Lemma H.8, the theorem follows.

$\square$

**Lemma H.2.** *Consider two sets of $m$ input points to an $n$-dimensional perceptron without bias, $\mathcal{U}$ and $\mathcal{V}$ with data matrices $U$ and $V$ respectively (see Definition H.1). If for all $i, j = 1, ..., m$, $(VV^T)_{ij}/\sqrt{(VV^T)_{ii}(VV^T)_{jj}} \geq (UU^T)_{ij}/\sqrt{(UU^T)_{ii}(UU^T)_{jj}}$, then every moment $\langle t^q \rangle$ of the distribution $P(T = t)$ is greater for the set of points $\mathcal{V}$ than the set of points $\mathcal{U}$.*

*Proof of Lemma H.2.* We write $T$ as:

$$T_s = \sum_{i=1}^m \mathbf{1}(\langle w, s_i \rangle)$$

$$T_u = \sum_{i=1}^m \mathbf{1}(\langle w, u_i \rangle)$$

The moments of the distribution $P(T_s = t)$ can be calculated by the following integral:

$$\langle t^q \rangle_{\mathbb{S}} = \int_S \sum_{i=0}^{m} \cdots \sum_{q=0}^{m} \mathbf{1}(\langle w, s_i \rangle) \times \cdots \times \mathbf{1}(\langle w, s_q \rangle) \, P(S)dS$$

Where $S = (S_1, \ldots, S_q)$. Taking the sum outside the integral,

$$= \sum_{i \ldots q} \int \mathbf{1}(\langle w, s_i \rangle) \times \cdots \times \mathbf{1}(\langle w, s_q \rangle) P(S)dS = \sum_{i \ldots q} P(\langle w, s_i \rangle > 0, \ldots, \langle w, s_q \rangle > 0)$$

The distribution for $\mathcal{U}$ is of the equivalent form. From corolloray Corollary H.7, we have that $P(\langle w, s_i \rangle > 0, \ldots, \langle w, s_q \rangle > 0) \leq P(\langle w, u_i \rangle > 0, \ldots, \langle w, u_q \rangle > 0)$. Thus we have Equation (22) for all $q$.

$$\langle t^q \rangle_{\mathbb{S}} < \langle t^q \rangle_{\mathcal{U}} \tag{22}$$

$\square$

**Lemma H.3.** *Consider an $2$-dimensional Gaussian random variable with mean $\mu$ and covariance $\Sigma$. If the correlation $\Sigma_{ij}/\sqrt{\Sigma_{ii}\Sigma_{jj}}$ increases, then*

$$P(x_i > 0, x_j > 0)$$

*increases*

*Proof.* We can write the non-centered orthant probability as a Gaussian integral

$$P(x_i > 0, x_j > 0) = \int_{\mathbb{R}_{\geq 0}} e^{-\frac{1}{2}(x-\mu)^T \Sigma^{-1}(x-\mu)} dx$$

Without loss of generality, we consider $\Sigma_{ii} = \Sigma_{jj} = 1$. Otherwise, we can rescale the variables $x$ and obtain a new mean vector.

The covariance matrix $\Sigma$ thus has eigenvalues $\lambda_+ = 1 + \Sigma_{ij}$ and $\lambda_- = 1 - \Sigma_{ij}$ with corresponding eigenvectors $(1, 1)$ and $(1, -1)$. We can rotate the axis so that $(1, 1)$ becomes $(1, 0)$. We can then rescale the $x$ axis by $1/\sqrt{\lambda_+}$ and the $y$ axis by $1/\sqrt{\lambda_-}$. The positive orthant becomes a cone $\mathcal{C}$ centered around the origin and with opening angle $\alpha$ given by

$$\tan \alpha = \sqrt{\frac{\lambda_+}{\lambda_-}},$$

which increases when $\Sigma_{ij}$ increases. The integral in polar coordinates becomes

$$\int_{\mathcal{C}} e^{-\frac{r^2}{2}} r d\theta dr.$$

Therefore, all that's left to show is that the range of $\theta$ for any $r$ increases when $\alpha$ increases. See Figure 10 for the illustration. Call $\gamma$ the angle between one boundary of the cone $\mathcal{C}$ and the position vector of the center of the Gaussian in the transformed coordinates. We can find the length of the chord between the two points of intersection between the circle and the boundary of the cone as the difference between the distances $c_-, c_+,$ of the segments OA and OB, respectively. Using the cosine angle formula, we find $c_\pm = d \cos \gamma \pm \sqrt{r^2 - d^2 \sin^2 \gamma}$, and the chord length is $\sqrt{r^2 - d^2 \sin^2 \gamma}$, which decreases as $\gamma$ increases. Furthermore, $\gamma$ increases as $\alpha$ increases. Using the same argument for the other boundary of the cone, concludes the proof.

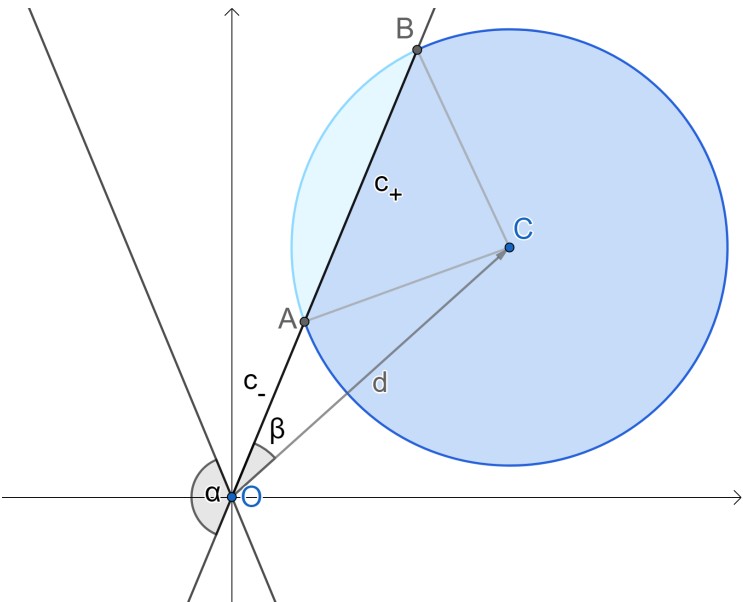

Figure 10: Transformed 2D Gaussian for proof in Lemma H.3.

$\square$

The following lemma shows that for a vector of $n$ Gaussian random variables, the probability of two variables being simultaneously greater than $0$, given that all other any fixed signs, increases if their correlation increases.

**Lemma H.4.** *Consider an $n$-dimensional Gaussian random variable with mean $0$ and covariance $\Sigma$, $x \sim \mathcal{N}(0, \Sigma)$. Consider, for any $\sigma \in \{-1, 1\}^{n-2}$, the following probability*

$$P_{00}^{ij} := P(x_i > 0, x_j > 0 | \forall k \notin \{i, j\} \sigma_k x_k > 0)$$

*If $\Sigma_{ij}/\sqrt{\Sigma_{ii}\Sigma_{jj}}$ increases, then $P_{00}^{ij}$ increases.*

*Proof.* We can write $P_{00}^{ij} = \mathbf{E}[P(x_i > 0, x_j > 0 | \hat{x}_{i,j})]$, where $\hat{x}_{i,j}$ is the vector of $x$ without the $i$th and $j$th elements, and the expectation is over the distribution of $\hat{x}_{i,j}$ conditioned on the condition $\forall k \notin i, j \sigma_k x_k > 0$.

The conditional distribution $P(x_i, x_j | \hat{x}_{i,j})$ is also a Gaussian with a, generally non-zero mean $\mu$, and a covariance matrix given by

$$\bar{\Sigma} = \begin{pmatrix} \Sigma_{ii} & \Sigma_{ji} \\ \Sigma_{ij} & \Sigma_{jj} \end{pmatrix} - \hat{\Sigma},$$

where $\hat{\Sigma}$ is independent of $\Sigma_{ij}$. This means that increasing the $\Sigma_{ji}$ (keeping $\Sigma_{ii}$ and $\Sigma_{jj}$ fixed) will increase the correlation in $\bar{\Sigma}$. Therefore, by Lemma H.3, $P(x_i > 0, x_j > 0 | \hat{x}_{i,j})$ increases, and thus $P_{00}^{ij}$ increases. $\square$

**Corollary H.5.** *An immediate consequence of Lemma H.4 is that $P(\forall i, x_i > 0) = P(x_i > 0, x_j > 0 | \forall k \notin \{i, j\} x_k > 0) P(\forall k \notin \{i, j\} x_k > 0)$ increases when $\Sigma_{ij}/\sqrt{\Sigma_{ii}\Sigma_{jj}}$ increases, as $P(\forall k \notin \{i, j\} x_k > 0)$ stays constant.*

**Lemma H.6.** *In the same setting as Lemma H.4, for covariance matrices $\Sigma$ and $\Sigma'$ of full rank, the following holds: for every $i, j$, $\Sigma_{ij} = \Sigma'_{ij}$ implies $P^{ij}_{K.00} = P^{ij}_{K'.00}$ and $\Sigma_{ij}/\sqrt{\Sigma_{ii}\Sigma_{jj}} > \Sigma'_{ij}/\sqrt{\Sigma'_{ii}\Sigma'_{jj}}$ implies $P^{ij}_{K.00} > P^{ij}_{K',00}$, where $P_K$ is the probability measure corresponding to covariance $K$.*

*Proof.* The set $\mathbb{S}$ of all symmetric positive definite matrices is an open convex subset of the set of all matrices. Therefore, it is path-connected. We can traverse the path between $\Sigma$ and $\Sigma'$ through a sequence of points such that the distance between point $x_i$ and $x_{i+1}$ is smaller than the radius of a ball centered around $x_i$ and contained in the set $\mathbb{S}$. Within this ball, one can move between $x_i$ and $x_{i+1}$ in coordinate steps that only change one element of the matrix. Therefore we can apply Lemma H.4 to each step of this path, which implies the theorem. □

**Corollary H.7.** *Consider any set of $m$ points $x^{(i)} \in \mathbb{R}^n$ with elements $x^{(i)}_j$. Let $X$ be the $m \times n$ data matrix with $X_{ij} = x^{(i)}_j$. For a weight vector $w \in \mathbb{R}^n$ let $y = Xw$ be the vector of real-valued outputs of an perceptron, with weights sampled from an isotropic Gaussian $\mathcal{N}(0, I_n)$. If the correlation between any two inputs increases, then $P(T = 0) = P([\sum_s \mathbf{1}(s)] = 0)$ increases*

*Proof.* The vector $y = \sum_i X_{i.} w_i$ is a sum of Gaussian vectors (with covariances of rank 1), and therefore is itself Gaussian, with a covariance given by $\Sigma = XX^T$. If the correlation product between any two inputs increases, then applying Lemma H.6 and Corollary H.5 at each step, the theorem follows. □

**Lemma H.8.** *If the uncentered moments of the distribution $P(T = t)$ increase, except for its mean (which is $2^{n-1}$), then $\langle H(t) \rangle$ increases.*

*Proof.* We consider the definition of the entropy, $H$, of a string (Definition 3.5). We define the first and second terms by $h_1 = (t/t_{max}) \ln(t/t_{max})$ and $h_2 = (1 - t/t_{max}) \ln(1 - t/t_{max})$. We taylor expand $h_2$ about $t = 0$:

$$(1 - t/t_{max}) \ln(1 - t/t_{max}) = (1 - t/t_{max}) \sum_k \frac{1}{k} \left( -\frac{t}{t_{max}} \right)^k = \sum_k \left( \frac{1}{k} + \frac{1}{k-1} \right) \left( \frac{t}{t_{max}} \right)^k$$

By symmetry, we see that $h_1(t) = h_2(t_{max} - t))$, and we can thus Taylor expand $h_1$ around $0$ $t_{max}$, to give:

$$\langle H(t) \rangle = \left\langle \sum_k a_k (t^k + (t_{max} - t)^k) \right\rangle = \left\langle \sum_k 2a_{2k} t^{2k} \right\rangle = \sum_k 2a_{2k} \langle t^{2k} \rangle$$

Because every $a_{2k} = \frac{1}{2k(2k-1)}$ is positive, we see that increasing every even moment increases the average entropy, $\langle H(t) \rangle$. □

## I  BIAS DISAPPEARS IN THE CHAOTIC REGIME

In the paper we have analyzed fully connected networks with ReLU activations, and find a general bias towards low entropy for a wide range of parameters. In a stimulating new study, Yang et al. ((Yang & Salman, 2019)) recently showed an example of a network using an erf activation function where bias disappeared with increasing number of layers.

Here we argue that the reason for this behaviour lies in the emergence of a chaotic regime, which does not occur for ReLU activations, but does for some other activation functions. In particular, the erf activation function is very similar to the tanh activation function used in an important series of recent papers that studied the propagation of correlations between hidden layer activations through neural networks, which they call "deep information propagation" ((Poole et al., 2016; Schoenholz et al., 2017; Lee et al., 2018)). They find that the activation function can have a dramatic impact on the behaviour of correlations. In particular, these papers show that for FCNs with tanh activation, there are two distinct parameter regimes in the asymptotic limit of infinite depth: One in which

inputs approach perfect correlation (the *ordered regime*), and one in which the inputs approach $0$ correlation (the *chaotic regime*). FCNs with ReLU activation do not appear to exhibit this chaotic regime, although they nevertheless have different dynamic regimes.

The particular example in (Yang & Salman, 2019) was for $\sigma_w = 4.0$, and $\sigma_b = 0.0$, a choice of hyperparameters that, for sufficient depth, lies deep in the chaotic regime. In this regime, inputs with initial correlations $> -1$ and $< 1$ will become asymptotically uncorrelated for sufficient depth, while initial correlations with equal to $\pm 1$ stay fixed, as we show below. The network is therefore equally likely to produce any odd function (on the $\{-1, 1\}^n$ Boolean hypercube).

To confirm our conjecture above, we performed experiments to calculate the bias of tanh networks in both the chaotic and ordered regime, which we show in Figure 11. We obtain the expected results: in the chaotic regime, the bias gets weaker with number of layers, while in the ordered regime, just as was found for the ReLU activation in the main text, the bias remains, and the trivial functions gain more probability. These experiment illustrate for tanh activation that in either the chaotic or ordered regime, the *a-priori* bias of the network becomes asymptotically degenerate with depth, either by becoming unbiased, or too biased.

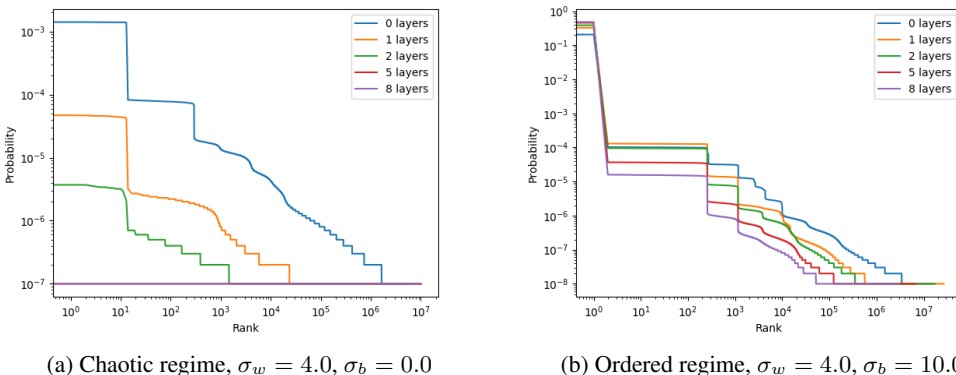

(a) Chaotic regime, $\sigma_w = 4.0$, $\sigma_b = 0.0$     (b) Ordered regime, $\sigma_w = 4.0$, $\sigma_b = 10.0$

Figure 11: Probability versus rank (ranked by probability) of different Boolean functions $\{-1, 1\}^7 \rightarrow \{-1, 1\}$ produced by a neural network with 1 hidden layer with tanh activation, and weights distributed by a Gaussian with different variance hyperparameters chosen to lie in the chaotic ($\sigma_b = 0.0$) and ordered ($\sigma_b = 10.0$) regimes.

We now explain that a simple extension of the analysis of (Poole et al., 2016) shows that, for the special case of $\sigma_b = 0.0$ and tanh activation function, initial correlations equal to $\pm 1$ stay fixed. This happens because the RHS in Equation 5. describing the propagation of correlation in (Poole et al., 2016) (which we replicate in Equation (23) below) is odd on the correlation $q_{12}^{l-1}$ (which represents the covariance between a pair of activations for inputs 1 and 2 at layer $l - 1$) when $\sigma_b = 0$. In addition to the fixed point they identified for the correlation being $+1$ there is therefore another unstable fixed point at $-1$.

$$q_{12}^l = \mathcal{C}\left(c_{12}^{l-1}, q_{11}^{l-1}, q_{22}^{l-1} | \sigma_w, \sigma_b\right) \equiv \sigma_w^2 \int \mathcal{D}z_1 \mathcal{D}z_2 \phi(u_1) \phi(u_2) + \sigma_b^2 \tag{23}$$

$$u_1 = \sqrt{q_{11}^{l-1}} z_1, u_2 = \sqrt{q_{22}^{l-1}} \left[ c_{12}^{l-1} z_1 + \sqrt{1 - \left(c_{12}^{l-1}\right)^2} z_2 \right], \tag{24}$$

where $c_{12}^l = q_{12}^l (q_{11}^l q_{22}^l)^{-1}$, and $z_1, z_2$ are independent standard Gaussian variables, and the $q_{11}$ and $q_{22}$ are the variances of the activations for input 1 and input 2, respectively. This fixed point at $-1$ ensures that points which are parallel but opposite (like opposite corners in the $\{-1, 1\}^n$ hypercube) will stay perfectly anti-correlated. This agrees with the expectation that the erf/tanh network with $\sigma_b = 0$ can only produce odd functions, as the activations are odd functions. However, any other pair of points becomes uncorrelated, and this explains why every (real valued) function, up to the oddness constraint, is equally likely. This also implies that every odd Boolean function is equally likely, as the region of function space satisfying the oddness constraint has the same shape within

every octant corresponding to an odd Boolean function. This is because the region in one octant is related to that on another octant by simply changing signs of elements of the function vector (a reflection transformation).

It will be interesting to study the effect of these different dynamic regimes, for different activation functions, on simplicity bias.

## J  BIAS TOWARDS LOW ENTROPY IN REALISTIC DATASETS AND ARCHITECTURES

In the main text we demonstrate bias towards low entropy only for the Perceptron, where we can prove certain results rigorously, and for a fully connected network (FCN). An obvious question is: does this bias persist for other architectures or data sets? In this Appendix we show that similar bias indeed occurs for more realistic datasets and architectures.

We perform experiments whereby we sample the parameters of a convolutional neural network (CNN) (with a single Boolean output), and evaluate the functions they obtain on subsets of the standard vision datasets MNIST or CIFAR10. Our results suggest that the bias towards low entropy (high class imbalance) is a generic property of ReLU-activated neural networks. This has in fact been pointed out previously, either empiricallyPage (2019), or for the limit of infinite depthLee et al. (2018). Extending our analytic results to these more complicated architectures and datasets is probably extremely challenging, but perhaps possible for some architectures, by analyzing the infinite-width Gaussian process (GP) limit.

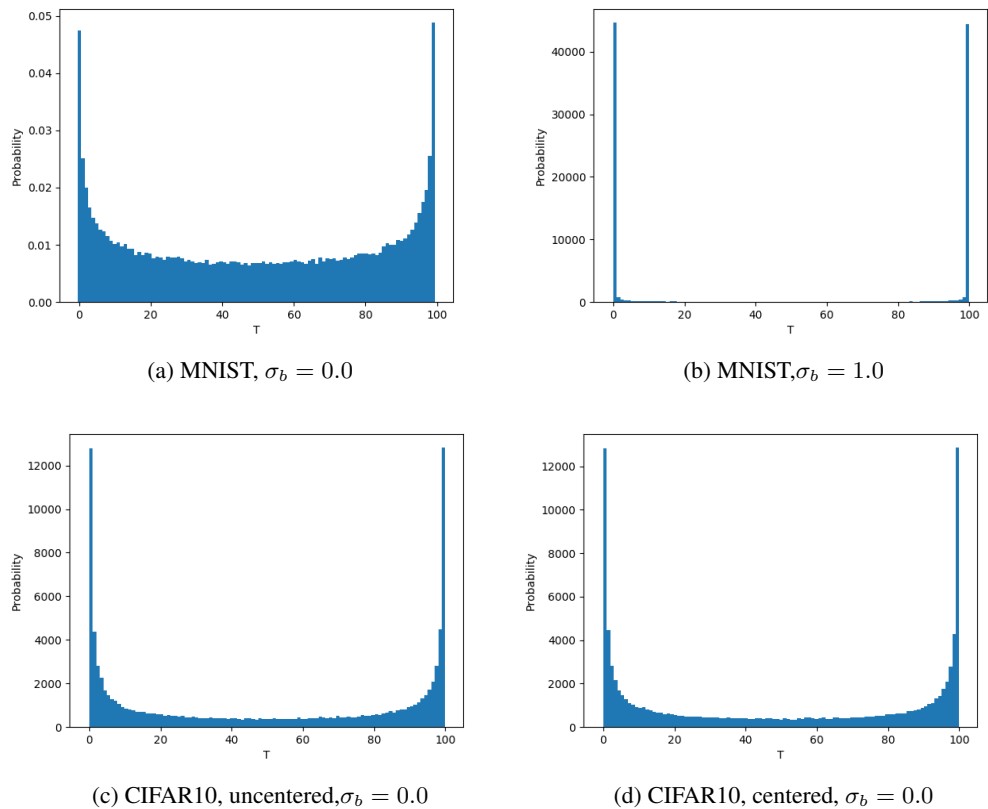

(a) MNIST, $\sigma_b = 0.0$

(b) MNIST, $\sigma_b = 1.0$

(c) CIFAR10, uncentered, $\sigma_b = 0.0$

(d) CIFAR10, centered, $\sigma_b = 0.0$

Figure 12: Probability of different values of $T$ for a CNN with $4$ layers, no pooling, and ReLU activations. The parameters were sampled $10^5$ times and the network was evaluated on a fixed random sample of $100$ images from MNIST or CIFAR10. The parameters were sampled i.i.d. from a Gaussian, with paramaters $\sigma_w = 1.0$, and $\sigma_b = 0.0, 1.0$. The input images from CIFAR10 where either left uncentered (with values in range $[0,1]$) (*uncentered*), or where centered by substracting the mean value of every pixel (*uncentered*).

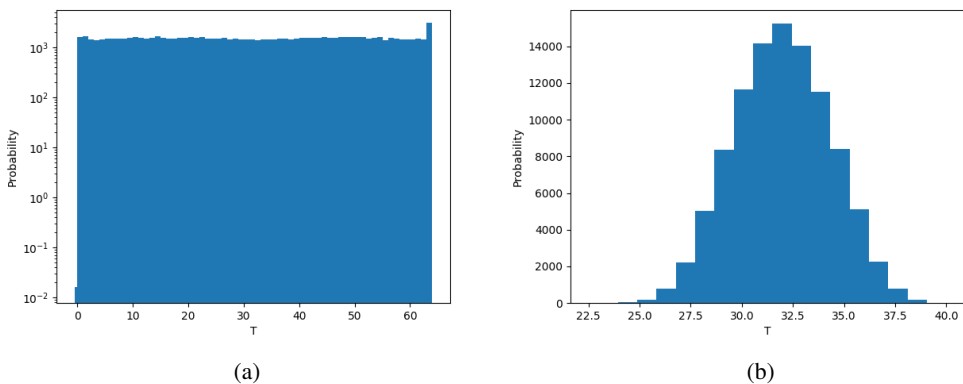

(a)

(b)

Figure 13: Probability of different values of $T$ for the perceptron with $n = 7$ input neurons and $\sigma_b = 0.0$. The parameters were sampled $10^5$ times and the perceptron was evaluated on a random subsample of size $64$ of either $\{0,1\}^n$ ((a)) or $\{-1,1\}^n$ ((b)). The weights were sampled i.i.d. from a Gaussian, with paramaters $\sigma_w = 1.0$.

## K    EFFECT OF THE BIAS TERM ON THE PERCEPTRON ON CENTERED DATA

Here we show that the bias towards low entropy is recovered for the perceptron on centered ($\{-1, 1\}^n$) inputs, when the bias term is increased.

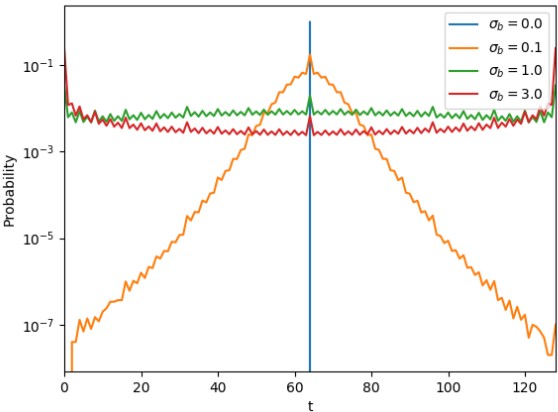

Figure 14: $\sigma_b = 1.0$

Figure 15: Probability of different values of $T$ for the perceptron evaluated on $\{-1, 1\}^n$ inputs and varying $\sigma_b = 0.0$. The parameters were sampled $10^7$ times. The weights were sampled i.i.d. from a Gaussian, with paramaters $\sigma_w = 1.0$.

## L    CONNECTION BETWEEN THE BIAS AT INITIALIZATION, INDUCTIVE BIAS AND GENERALIZATION

In this appendix we describe more formally the connection between the distribution over functions at initialization and the inductive biases, when training with an optimizer like SGD, and the link to generalisation.

Following the ideas in (Valle-Pérez et al., 2018), we will first formally introduce an exact Bayesian classifier for classification.

Let $P(\theta)$ be a prior distribution over parameters, which induces a prior distribution over Boolean functions given by $P(f) := P(\Theta_f)$, where we define $\Theta_f$ to be the set of parameter values that produce the function $f$. We assume a 0-1 likelihood $P(D|f)$ for data $D = \{(x_i, y_i)\}_{i=1}^m$, defined as

$$P(D|f) = \begin{cases} 1 \text{ if } \forall i f(x_i) = y_i \\ 0 \text{ otherwise} \end{cases}$$

The likelihood on parameter space is defined as $P(D|\theta) := P(D|f_\theta)$, where $f_\theta$ is the function produced by parameter $\theta$. The Bayesian posterior on function space is then

$$P(f|D) = \frac{P(D|f)P(f)}{P(D)},$$

where $P(D) = \sum_f P(D|f)P(f)$ is the *marginal likelihood* of the data. In parameter space, the posterior is just $P(\theta|D) = \frac{P(D|\theta)P(\theta)}{P(D)}$

For an exact Bayesian algorithm, like the one described above, the inductive bias (the preference of some functions versus others, for a given dataset), is fully encoded in the prior $P(f)$.

The connection with generalisation for the Bayesian learner described above can be expressed through the PAC-Bayes theorem (McAllester, 1999; Valle-Pérez et al., 2018) which gives bounds on

the expected generalization performance that depend on the marginal likelihood, which is just the probability of the labelling of the data under the prior distribution. In this picture, the distribution $P(f)$ at initialization determines the full inductive bias, but only to extent to which the training algorithm approximates the Bayesian posterior, with $P(f)$ as its prior.

In (Valle-Pérez et al., 2018) it was shown that PAC-Bayes bounds work well for an FCN and a CNN trained on CIFAR, and also for an FCN on Boolean data. The success of these bounds is indirect evidence that the training algorithm (here different variants of SGD) is indeed behaving somewhat like a Bayesian sampler.

We are aware that the claim in the paragraph above may be controversial, given that there is also a literature arguing that SGD itself is an important source of the inductive bias of deep learning. So it is important to also find direct evidence that that stochastic gradient descent (SGD) approximates the Bayesian posterior with prior $P(f)$.

In(Valle-Pérez et al., 2018), Fig 4, the probability $P(f)$ was compared to the probability that two variants of SGD generate functions, for the Boolean system. Good agreement was found, suggesting that SGD indeed samples functions with a probability close to $P(f)$, at least on the log scale and for these problems.

Here we provide further evidence for this behaviour in Figure 16, where we compare the probabilities of finding different Boolean functions upon training a fully connected neural network with SGD to learn a relatively simple Boolean function versus the probabilities of obtaining those functions by randomly sampling parameters until they fit the data (which we refer to as approximate Bayesian inference, or ABI). We see that most functions in the sample show very similar probabilities to be obtained by either SGD or ABI.

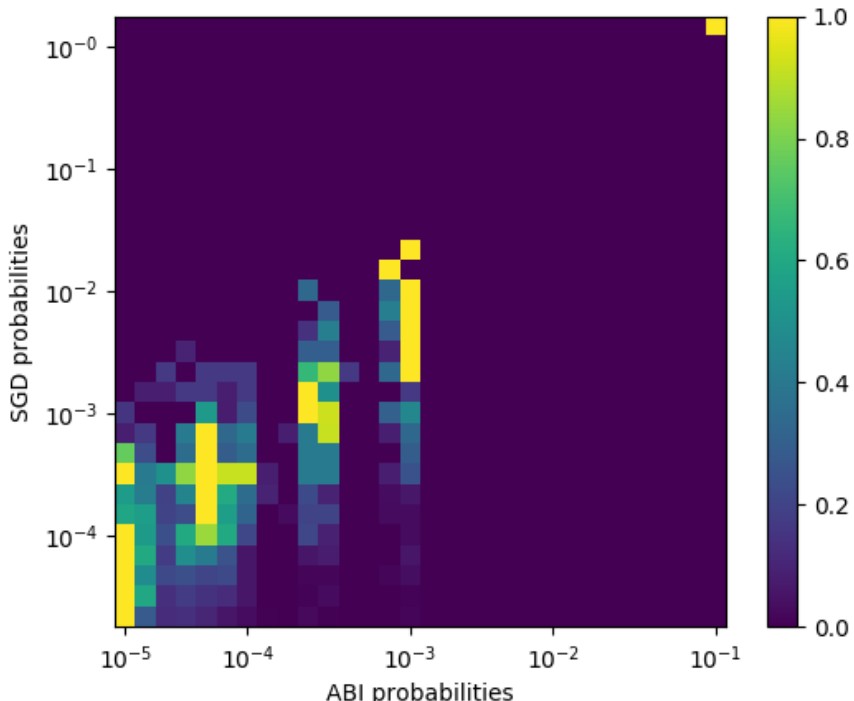

Figure 16: Probabilities of finding functions by SGD versus by randomly sampling parameters (ABI), conditioned on 100% training accuracy, for a training set of size 32 for learning a Boolean function on $\{0,1\}^7$ with a fully connected neural network with layer widths (7,40,40,1). Both the parameter sampling and SGD initialization where i.i.d. Gaussians with weight variances of $1/(\text{layer width})$, and bias variances of $1$. SGD used a batch size of 8, and cross-entropy loss, and was stopped as soon as 100% accuracy was reached. Histogram shows number of functions on each bin, and is normalized row-wise, so that a value of 1.0 corresponds to the maximum number of functions in the row, and 0.0 corresponds to 0.0 functions. These are mapped to colors from yellow (1.0) to purple (0.0) as shown in the colorbar.

Although we don't yet have a formal theoretical explanation of this correlation, we can make the following heuristic argument: The parameter-function map (defined in Section 3), is hugely biased, typically over many orders of magnitude. The basins of attraction of the functions are also likely to vary over many orders of magnitude in size, so that SGD is much more likely to find functions with larger $P(f)$, which have larger basins, than functions with smaller $P(f)$ (See also (Wu et al., 2017)). This argument would be sufficient to show that the bias upon initialisation (or equivalently upon uniform random sampling of parameters) has an important effect on the functions that SGD finds. However, empirically we find a stronger result, which is that SGD approximately samples functions with a probability directly proportional to $P(f)$. This agreement suggests another stronger ansatz: As long as SGD samples the parameters more or less uniformly (within the region of likelihood 1), then on function space the functions are samples with probabilities of the same orders of magnitude as $P(f|D)$.

Finally, preliminary results on MNIST and CIFAR data sets (using the GP approximation) exhibit a similar scaling to what is observed in Figure 16. We are currently working further on this important and complex question of how SGD samples functions.

## M  EFFECT OF BIAS ON LEARNING

The effect of inductive biases such as the ones we discuss on learning is a vast and open research project. In particular, we expect that bias towards low entropy should play a bigger role when trying to learn class-imbalanced data. In class-imbalanced problems, one is often interested in quantities beyond the raw test accuracy. For example, the *sensitivity* (the fraction of test-set misclassifications on the rare class), and the *specificity* (the fraction of test-set misclassifications on the common class). Furthermore, the use of tricks, like oversampling the rare class during optimisation, makes the formal analysis of practical cases even more challenging. In this section, we show preliminary results illustrating some of the effects that entropy bias in $P(f)$ can have on learning.

In the experiments, we shift the bias term of the output neuron $b \mapsto b' = b + \text{shift}$. This causes the functions at initialisation to have distributions $P(T)$ which may be biased towards higher or lower values of $T$ (mapping most inputs to 0 or 1). We measure the average $\langle T \rangle$ for different choices of the shift hyperparameter, and train the network, using SGD, on a class-imbalanced dataset. Here we perform this experiment on a CNN with 4 layers (and no pooling), and a FCN with 1 layer. The dataset consists of a sample of either MNIST (10 classes) or balanced EMNIST (47 classes), where a single class is labelled as 0, and the rest are labelled as 1.

We see in Figure 17 that values of the shift hyperparameter producing large values $\langle T \rangle$ give higher test accuracy. We suggest that this could be because the new parameterisation induced by the shifted bias term[10], causes the inductive bias is more "atuned" to the correct target function (which has a class imbalanced of 1:10 and 1:47, respectively for MNIST and balanced EMNIST). However, the detailed shape of the curve seems architecture and data dependent. For instance, for the FCN (with ReLU or tanh activation) trained on MNIST we find a clear accuracy peak around the true value of $\langle T \rangle$, but not for the others.

We also measured the sensitivity and 99% specificity (found by finding the smallest threshold value of the sigmoid unit, giving 99% specificity). Sensitivity seems to follow a less clear pattern. For two of the experiments, it followed the accuracy quite closely, while for FCN trained on MNIST it peaked around a positive value of the ship hyperparameter.

These results suggest that looking at architectural changes that affect the entropy bias of $P(f)$ can have a significant effect on learning performance for class-imbalanced tasks. For example, for the tanh-activated FCN, increasing the shift hyperparameter above 0 improves both the accuracy and sensitivity significantly. As we mentioned at the beginning, understanding the full effect of entropy bias, and other biases on $P(f)$, on learning is a large research project. In this paper, we focused on explaining properties of $P(f)$, while only showing preliminary results on the effect of these properties on learning.

---

[10]Note that the shifted bias reparameterisation is equivalent to a shifted *initialization* in the original parameterisation. For more complex reparameterisation, this may not be the case

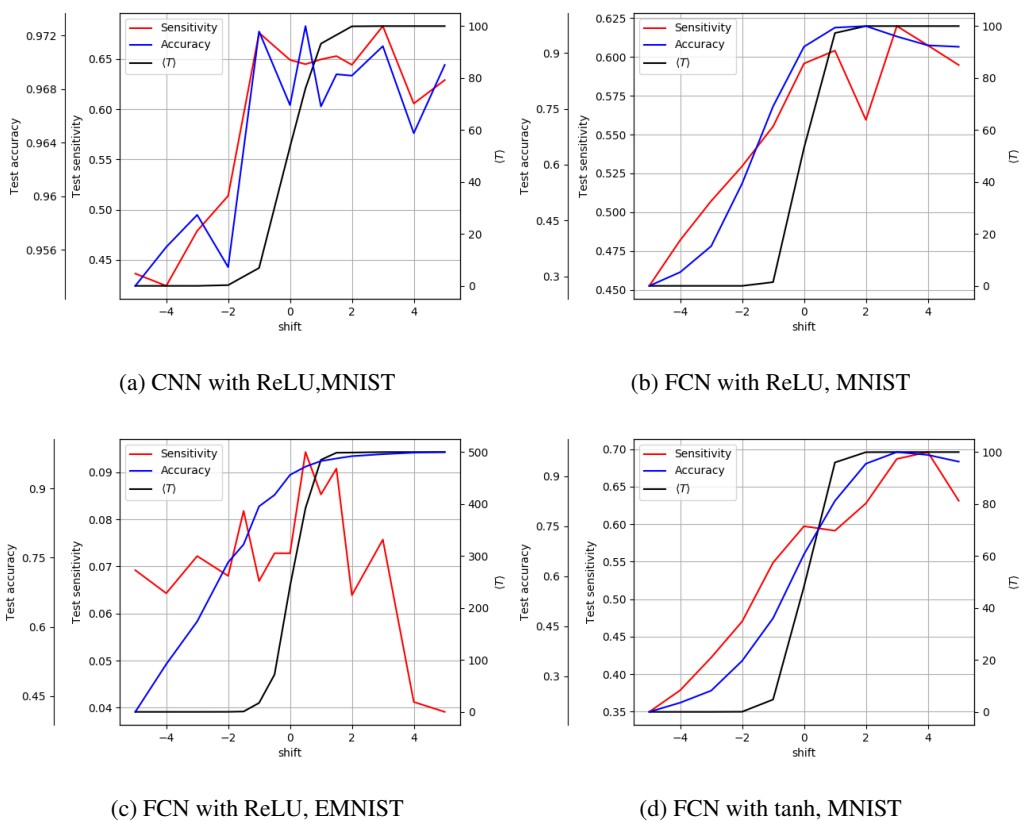

(a) CNN with ReLU,MNIST

(b) FCN with ReLU, MNIST

(c) FCN with ReLU, EMNIST

(d) FCN with tanh, MNIST

Figure 17: Test accuracy, sensitivity (at 99% specificity), and $\langle T \rangle$ versus the shift hyperparameter (shift of the bias term in the last layer), for different architectures and datasets. The networks were trained with SGD (batch size 32, cross entropy loss), until reaching 100% training accuracy. Accuracies and sensitivities are averages over 10 SGD runs with random initializations ($\sigma_b = 0.0$, $\sigma_w = 1.0$). The dataset had size 100 for MNIST, and 500 for EMNIST. Images are centered, so pixel values have their mean over the whole dataset substracted. The values of $\langle T \rangle$ are estimated from 100 random parameter samples and evaluating on the same data we train the network on.

