# OpenReview forum: "Neural networks are a priori biased towards Boolean functions with low entropy"
_ICLR.cc/2020/Conference — Reject_

### Official Review · AnonReviewer3 · 2019-10-23
**Official Blind Review #3**

**Rating:** 6

**Review:**

This paper studies the a-priori bias of a feed-forward neural network when the weights are initialised uniformly at random and independent of the network architecture. The paper claims that this initialisation leads to biases towards low entropy functions when the input and output are binary values.

The paper starts with a single layer perceptron without the bias term, and generalises the analysis to networks with multiple hidden layers and ReLU activations. The proposed approach seems rigorous, but I have a hard time to follow the paper as many of the important results are presented in appendix. In addition, the analysis is based on a feed-forward neural network with binary inputs and a single binary output, it is not clear whether these results can be generalised to architectures of practical importance such as convolutional/recurrent neural networks. Overall an interesting piece of work that contributes to the understanding of deep neural networks.


Minor comments:

Section 4.2:
"as as predicted by Eq. 1" -> remove duplicated "as"

Section 5.2:
"some interesting recent work" -> "Some ..."
"produce.At first sight" -> add a space before "At"
"If there is not bias" -> "If there is no bias"

**Experience Assessment:**

I do not know much about this area.

**Review Assessment: Checking Correctness Of Derivations And Theory:**

I assessed the sensibility of the derivations and theory.

**Review Assessment: Checking Correctness Of Experiments:**

N/A

**Review Assessment: Thoroughness In Paper Reading:**

I read the paper at least twice and used my best judgement in assessing the paper.

---

> ### Author Response · Authors · 2019-11-14
> **Response to reviewer 3**
>
> We thank the referee for the positive assessment of our work.
>
> There is one important critique:  "it is not clear whether these results can be generalised to architectures of practical importance such as convolutional/recurrent neural networks"
>
> Stimulated by this question,  we have now performed new experiments on a convolutional neural network (CNN) on the MNIST database, as well as a CNN on CIFAR-10, and also on the perceptron with sub-samples of the hypercube.  All these different architectures and data show very similar anti-entropy bias as found for the simpler architectures and datasets studied in the main text.    Full results can be found in SI section J, confirming that the results we find here carry on to other architectures and datasets.
>
> The referee also feels that having many key results  in the appendices makes the paper hard to follow.  We are aware of this problem. Unfortunately, many of the proofs are rather long, and due to space constraints, we unfortunately  had to stick them in the Appendix.  Short of a larger page limit, we don't think we can easily fix this problem.
>
> We have fixed the minor comments.

---

### Official Review · AnonReviewer1 · 2019-10-23
**Official Blind Review #1**

**Rating:** 6

**Review:**

The authors study the behavior of simple neural networks at initializations. Particularly, the authors show that at initializations, neural networks tend to be functions with high class imbalance.
Further, the authors show that how such conclusion would be reached with or without a bias term, with different number of hidden layers, with change of activation functions.
The work is fairly interesting, yet the motivation is less clear.
The conclusion that study of such initializations can help understand the generalization power is not convincing.
Despite that neural networks at initializations are biased towards low entropy functions, it’s not clear with training on a dataset with an optimizer, how much we can conclude about the generalization power.
Overall the paper is well written.

Below are some more detailed comments:
1) In the Introduction and the first paragraph of Section 2, the authors motivate by describing how important it is to understand the inductive biases. Yet the study is about the behavior of a random initialization. It would be nice to tie these two better; or motivate from another angle, other than the inductive bias. To my reading, the sentence “what the inductive biases … are, and how they arise” is not well supported because I still don’t follow what the inductive biases are from reading the paper except that a random initialization likely be low entropy functions.

2) In Figure 3(a), 4(b,c,d), there is a spike at the mid-point of t. Though not as high as the extreme points, this is contradictory to the main conclusion. It would nice to add discussions.

3) It would be nice to add experiments to study how such bias at initializations would impact the model training.


**Experience Assessment:**

I have read many papers in this area.

**Review Assessment: Checking Correctness Of Derivations And Theory:**

I did not assess the derivations or theory.

**Review Assessment: Checking Correctness Of Experiments:**

I assessed the sensibility of the experiments.

**Review Assessment: Thoroughness In Paper Reading:**

I read the paper at least twice and used my best judgement in assessing the paper.

---

> ### Author Response · Authors · 2019-11-14
> **Response to reviewer 1**
>
> We thank the referee for a careful reading of our paper and for the positive assessments.
>
> The referee makes one major critique: "The conclusion that study of such initializations can help understand the generalization power is not convincing. Despite that neural networks at initializations are biased towards low entropy functions, it’s not clear with training on a dataset with an optimizer, how much we can conclude about the generalization power."
>
> We are grateful for the opportunity to clarify this important question: It is true that if optimisation were to scramble all the bias at initialisation, then our results would not be that relevant to generalisation.
>
> However, we will argue that much of this bias persists upon optimisation, although this is a topic of active research.  In particular, we have added new new Appendix L, where we explain the connection between generalisation,  the prior over functions at initialization and the inductive bias after training. We now also clarify these connections better in the 2nd paragraph in the introduction.
>
> In Appendix L, we  demonstrate empirically for a simple FCN and Boolean data that the probability that a function is chosen with SGD, when reaching 0 training error on the data, is highly correlated to the probability that the function is chosen by random sampling of weights (equivalent to initialisation), conditioned on 0 training error. We have further evidence that this trend holds for more complex systems where we use the Gaussian process (GP) approximation to calculate the probabilities upon random sampling of data, which we are currently writing up for a separate publication.  We have a rough blog post  summarising this data (going well beyond Appendix L) - see it at https://bit.ly/2qSPXnH section "what does SGD do?".   While further study is needed to more comprehensively explore these complex issues,  so far all our data very strongly points towards the bias at initialisation strongly carrying on through optimisation, and thus on to generalisation.
>
> Our arguments above suggest that the inductive bias of a DNN should be mainly determined by the prior P(f) (defined by randomly sampling weights). Furthermore, the prior can be used to predict the generalization performance, using the PAC-Bayesian theorem (as done in https://arxiv.org/abs/1805.08522). Intuitively, if a DNN is highly biased towards a certain type of function, then this bias should improve generalisation on data that is described by such functions.   Conversely, if the bias in the DNN is opposite to that of the data, generalisation  performance will suffer.  This is one way to understand why the generalisation depends on what the data is like.
>
> We have also added a new Appendix M where we show some results on the effect of the bias in P(T) -- for example, towards producing mostly 1s (high T) or mostly 0s (low T) -- on learning class imbalanced data. These are preliminary and merit further research, but should offer some evidence that the prior P(f) has strong effects on generalization, as predicted by arguments in Appendix L.
>
> Finally, we note that the correlation between the bias at initialisation and upon training was discussed at some length at https://arxiv.org/abs/1805.08522. Our paper sheds more light on this issue, but it is not yet settled.  We hope the current paper will stimulate new work in this field.
>
> -- Response to more detailed comments
> 1)  The main response to this question can be found above.   We agree with the referee  that we could be clearer in the introduction and in section 2, and we have made improvements there.   In particular, we added Appendices K and M which explain the connections in more detail.
>
> 2).  For {0,1}^n, the peaks are largest for numbers of t’s that have few 1s in their binary representation (e.g. powers of 2). These effects arise for a finite bias term, for which we don’t have (yet) analytic results. We are currently working on exactly why these peaks occur, and  believe that the peaks are related to increased expressivity for functions in these classes, rather than increases in the probability of individual functions in these classes.  But we have no definitive results yet. Note that these peaks rapidly disappear upon addition of further layers, and so such features are less important for practical DNNs.  Also, in response to referee 4, we have added further discussion on the peak at the midpoint that appears for fully centered data  where symmetries mean that only functions with max entropy, and no others, can be expressed.  This is a rather singular case (the effect goes away with a bias term, or with adding layers or for other data), and is likely specific to the symmetry of this data set for the perceptron, and  therefore different in character from the peaks seen in Fig 3 for example.
>
> 3) We thank the referee for this suggestion.  Our results on the effect of training can be found in the new Appendix M, supported by arguments in Appendix L.

---

### Official Review · AnonReviewer4 · 2019-11-09
**Official Blind Review #4**

**Rating:** 3

**Review:**

The topic of the paper is the inductive bias of neural networks. The authors study a simple model, namely a perceptron with no bias term viewed as a mapping from {0,1}^n->{0,1}. They show that initializing weights with a distribution that is symmetric under coordinate sign flips corresponds to an initialization in function space that is biased towards low-entropy functions. They also exhibit empirical evidence that by adding a bias term, or by using multiple layers, this tendency towards low entropy appears to increase. They also prove a bound on the minimal size of a network in order for it to represent all boolean functions. Finally, they prove a result that suggests that for ReLU networks with infinite widths the bias towards low-entropy function does indeed increase with depth.

My main concern regarding this paper is that the claim in the title and the statement of Theorem 4.1 seem to rely crucially on the fact that the functions are viewed with input as {0,1}^n. The origin of the "simplicity", in the basic case that the authors address (perceptrons with no bias) appears to be a consequence that hyperplanes through the origin are quite likely to classify input points in {0,1}^n similarly. If one switches to a symmetric domain, for example {-1,1}, the effect in this setting completely disappears. The authors actually mention this in Section 5, noting that the expressivity of the perceptron is much lower for centered inputs. However, this to me suggests that Theorem 4.1 is not capturing any significant aspect of neural networks (in fact, the statement is a property of how linear hyperplanes to separate {0,1}^n, not neural networks). I may be mistaken, but I would like the authors to clarify this point.

Another concern is related to Theorem 5.5. This might be a more substantial result, but it is difficult to interpret and its implications are not discussed. Understanding the effect of depth on the "simplicity bias" seems to me an important problem, but for some reason Theorem 5.5 (which deals with deep neural networks rather than linear perceptrons) is emphasized much less than Theorem 4.1. Why is this the case?

The paper is well-written but not always very clear. In particular, notation is not always defined and the authors use on notions from complexity theory that are not introduced (e.g., Lepel-Ziv complexity).

Other comments:

* Is the set F_t is defined as set of all functions with assigned \mathcal T, but later this seems to be restricted to the functions expressible by a network/perceptron.
* Definition 3.5: this defined the entropy H(f) of a function but then write H(p). It should probably be H(f) = -plog p - (1-p)log(1-p) where p = \mathcal T(f), right?
* Definition 3.6: some context or references for this definition could be useful.
* Regarding the fact that functions in F_t are not uniform, shouldn't the distribution be the same for isotropic weight distributions? Assuming the distribution of w/|w| is uniform on the sphere, a more precise description of  P(f) seems possible
* Section 4.3: what is the "rank" in this setting? Isn't the parameter a vector w in R^n?
* Some typos: Definition 3.1 w_l \in R^{n_{l+1}}, ",." in the beginning of Section 5, several in the last paragraph of Section 5.

**Experience Assessment:**

I do not know much about this area.

**Review Assessment: Checking Correctness Of Derivations And Theory:**

I assessed the sensibility of the derivations and theory.

**Review Assessment: Checking Correctness Of Experiments:**

N/A

**Review Assessment: Thoroughness In Paper Reading:**

I read the paper at least twice and used my best judgement in assessing the paper.

---

> ### Author Response · Authors · 2019-11-14
> **Response to reviewer 4 (part 1)**
>
> We thank the reviewer for the useful comments.
> The main concern is that theorem 4.1 relies on the assumption that the inputs are uniformly distributed on {0,1}^n, and so may not be capturing a general phenomenon.  We agree with the referee that this point was not clarified as much as it could be in the main text.
> It is also true that the assumption in our theorem that the inputs are {0,1}^n is strong (although note that our theorem can be trivially extended to apply to the corners of any linear transformation of {0,1}^n corresponding to stretching axes).  In our defence, analytic results in this area are rare and difficult to achieve, and so we needed some simplifying assumptions.  Nevertheless, we show empirical evidence that the effect of bias towards low entropy is much more general than what we are currently able to prove. For instance, we show ``anti-entropy’’ bias in all these cases:
> • zero or nonzero bias for {0,1}^n inputs
> • nonzero bias for {-1,1}^n inputs (see Figure 15 in Appendix K)
> • sufficiently deep FCNs, for any input distribution (from Theorem 5.5). And also FCN of *any* depth on {0,1}^n, (Theorem 4.1 + Theorem 5.5)
> • 4-layer CNN with zero or nonzero bias, for a sample of MNIST (added in Appendix J)
> • 4-layer CNN with zero bias, for an uncentered and a centered sample of CIFAR-10 (added in Appendix J)
> • a perceptron using random subsets of {0,1}^n also generates  a distribution P(T) that is  close to uniform. (added in Appendix J)
>
> The main counterexample we could find is for {-1,1}^n  on the perceptron without bias. Note that this case is odd.  Only functions with maximum entropy can be generated, which corresponds to a large reduction in expressivity.  Even so, there is still simplicity bias within this set - in other words functions with low Lempel Ziv (LZ) complexity have much higher probability than functions with high LZ complexity (Figure 6b in Appendix D). Upon addition of layers, or for a large enough bias term, the system reverts back to the generic anti-entropy bias behaviour seen for all the other systems (Figures 4d and Figure 15).  So we argue that this particular case is a singular one, brought about by particular symmetries of the perceptron with zero bias term.  The fact that there is a (potentially interesting) exception does not negate our main result which we find holds much more generally.
>
> The reviewer also feels that Theorem 5.5 was underemphasized.   It is true that this theorem holds for more general architectures and datasets than Theorem 4.1 does.  To clarify, it shows that the bias towards low entropy increases monotonically with increasing depth, rather than asymptotically in the limit of infinite depth, as previous results have suggested. It also uses a more quantitative measure of the bias, via the average entropy.  However, it doesn’t tell you how anti-entropy biased a network is, it just tells you that if you make it deeper, it will become more anti-entropy biased.   Theorem 4.1 is complementary to this, because it gives some conditions on the data that guarantee anti-entropy bias.
>
> Although we do mention Theorem 5.5 in the abstract, we agree that we should have emphasised this more, and so  have now done so at a few key parts of the paper. For example, in the discussion, we now refer to it as one of our main results.
>
> We apologize for parts of the paper that have undefined notation, that should all be fixed now.
> A full description of the Lempel Ziv complexity measure can be found in  Valle-Perez et al.,  https://arxiv.org/abs/1805.08522.

---

> ### Author Response · Authors · 2019-11-14
> **Response to reviewer 4 (part 2; minor comments)**
>
> In response to "Other comments":
>
> * Is the set F_t is defined as set of all functions with assigned \mathcal T, but later this seems to be restricted to the functions expressible by a network/perceptron.
> We have fixed this to clarify that F_t is all functions expressible by a given model with assigned \mathcal T.
>
> * Definition 3.5: this defined the entropy H(f) of a function but then write H(p). It should probably be H(f) = -plog p - (1-p)log(1-p) where p = \mathcal T(f), right?
> This has been fixed.
>
> * Definition 3.6: some context or references for this definition could be useful.
> This is a measure of the shortest description of a Boolean function.  We have added a paragraph explaining the definition with more context.
>
> * Regarding the fact that functions in F_t are not uniform, shouldn't the distribution be the same for isotropic weight distributions? Assuming the distribution of w/|w| is uniform on the sphere, a more precise description of  P(f) seems possible
> If we have understood this point correctly, then one response is that we show section 4 and in the Appendix E that while Gaussian (weight distributions  w/|w| is uniform on the sphere) and uniform weight distributions on [-1,1]^n give the same  2^(-n) probability for each set of functions with a given t, the individual functions themselves can have different probabilities.  We can prove probabilities for some functions directly, and they are not the same.  Expressivity may also not be the same for different isotropic weight distributions.
>
> * Section 4.3: what is the "rank" in this setting? Isn't the parameter a vector w in R^n?
> To clarify this point we have now added to section 4.3 the sentence : To define the rank,  we order the functions by decreasing probability, and then the rank of a function f is the index off under this ordering (so the most probable function has rank 1, the second rank 2 and so on).
>
> * Some typos: Definition 3.1 w_l \in R^{n_{l+1}}, ",." in the beginning of Section 5, several in the last paragraph of Section 5.
>   -	thanks, these have been fixed.

---

### Decision · Program_Chairs · 2019-12-19

**Decision:**

Reject

**Comment:**

This article studies the inductive bias in a simple binary perceptron without bias, showing that if the weight vector has a symmetric distribution, then the cardinality of the support of the represented function is uniform on 0,...,2^n-1. Since the number of possible functions with support of extreme cardinality values is smaller, the result is interpreted as a bias towards such functions. Further results and experiments are presented. The reviewers found this work interesting and mentioned that it contributes to the understanding of neural networks. However, they also expressed concerns about the contribution relying crucially on 0/1 variables, and that for example with -1/1 the effect would disappear, implying that the result might not be capturing a significant aspect of neural networks. Another concern was whether the results could be generalised to other architectures. The authors agreed that this is indeed a crucial part of the analysis, and for the moment pointed at empirical evidence for the appearance of this effect in other cases. The reviewers also mentioned that the motivation was not very clear, that some of the derivations were difficult to follow (with many results presented in the appendix), and that the interpretation and implications were not sufficiently discussed (in particular, in relation to generalization, missing a more detailed discussion of training). This is a good contribution and the revision made important improvements on the points mentioned above, but not quite reaching the bar.